# Protective effect of TCR-mediated MAIT cell activation during experimental autoimmune encephalomyelitis

Mark Walkenhorst [1], Jana K. Sonner [1], Nina Meurs [1], Jan Broder Engler [1], Simone Bauer[1], Ingo Winschel [1], Marcel S. Woo [1], Lukas Raich[1], Iris Winkler [1], Vanessa Vieira[1], Lisa Unger[1], Gabriela Salinas[2], Olivier Lantz [3], Manuel A. Friese [1,4] ✉ & Anne Willing [1,4] ✉

Mucosal-associated invariant T (MAIT) cells express semi-invariant T cell receptors (TCR) for recognizing bacterial and yeast antigens derived from riboflavin metabolites presented on the non-polymorphic MHC class I-related protein 1 (MR1). Neuroinflammation in multiple sclerosis (MS) is likely initiated by autoreactive T cells and perpetuated by infiltration of additional immune cells, but the precise role of MAIT cells in MS pathogenesis remains unknown. Here, we use experimental autoimmune encephalomyelitis (EAE), a mouse model of MS, and find an accumulation of MAIT cells in the inflamed central nervous system (CNS) enriched for MAIT17 (RORγt⁺) and MAIT1/17 (T-bet⁺RORγt⁺) subsets with inflammatory and protective features. Results from transcriptome profiling and Nur77GFP reporter mice show that these CNS MAIT cells are activated via cytokines and TCR. Blocking TCR activation with an anti-MR1 antibody exacerbates EAE, whereas enhancing TCR activation with the cognate antigen, 5-(2-oxopropylideneamino)−6-D-ribitylaminouracil, ameliorates EAE severity, potentially via the induction of amphiregulin (AREG). In summary, our findings suggest that TCR-mediated MAIT cell activation is protective in CNS inflammation, likely involving an induction of AREG.

Multiple sclerosis (MS) is a chronic inflammatory disease of the central nervous system (CNS), afflicting more than 2.8 million people worldwide. Pathological hallmarks of MS include inflammatory lesions with demyelination and neuro-axonal degeneration[1]. The disease mostly starts in young adulthood with diverse and progressive neurological deficits. MS is thought to be initiated by a breakdown of immune tolerance mediated by autoantigen-specific T cells followed by infiltration of additional immune cells from the periphery and activation of CNS-intrinsic immune cells. Evidence for an interplay of lymphocytic and myeloid immune cells in the pathogenesis, apart from classical

autoreactive or regulatory T cells, has recently been assembled and represents a major focus of MS research[2].

In the MS animal model, experimental autoimmune encephalomyelitis (EAE), auto-aggressive T cells directed against myelin antigens are expanded by immunization and infiltrate the CNS leading to inflammation further attracting and activating other immune cells. The majority of mice develop classical EAE symptoms with an ascending paralysis from the tail to the hindlimbs correlating with CNS inflammation[3]. While both, Th1 and Th17 cells, can induce EAE, pathogenic Th17 cells co-expressing the transcription factors RORγt

[1]Institute of Neuroimmunology and Multiple Sclerosis, University Medical Center Hamburg-Eppendorf, Hamburg, Germany. [2]NGS-Integrative Genomics Core Unit, Institute of Pathology, University Medical Center Göttingen, Göttingen, Germany. [3]Institut National de la Santé et de la Recherche Médicale U932, PSL University, Institut Curie, Paris, France. [4]These authors contributed equally: Manuel A. Friese, Anne Willing. ✉e-mail: manuel.friese@zmnh.uni-hamburg.de; anne.willing@zmnh.uni-hamburg.de

and T-bet and producing the cytokines IFN-γ, GM-CSF and IL-17A are considered to be the main drivers of EAE progression[4,5]. The role of unconventional T cells, including mucosal-associated invariant T (MAIT) cells, in MS and EAE is less well understood.

MAIT cells, an innate-like T cell population, have a semi-invariant αβ T cell receptor (TCR) consisting of TRAV1-2 joined to TRAJ33 and paired with a limited repertoire of Vβ chains recognizing derivatives of metabolites of the riboflavin (vitamin B2) pathway of yeast and bacteria presented by the non-polymorphic MHC class I-related protein 1 (MR1)[6–8]. Until now, two specific natural antigens derived from the riboflavin pathway have been identified to strongly activate MAIT cells via their TCR, 5-(2-oxopropylideneamino)−6-D-ribitylaminouracil (5-OP-RU) and 5-(2-oxoethylideneamino)−6-D-ribitylaminouracil (5-OE-RU)[9]. Recently, cholic acid 7-sulfate (CA7S), a sulfated bile acid, as a first endogenous antigen has been reported to activate MAIT cells in a MR1-dependent manner[10]. In addition, MAIT cells can get activated in a TCR-independent, innate-like manner by inflammatory cytokines including IL-12 and IL-18[11]. Thereby, MAIT cells participate not only in defence against bacterial infections[12,13], but also against viral infections[14,15] in mice and humans.

MAIT cell effector functions include secretion of inflammatory cytokines, such as GM-CSF, IFN-γ, TNF and IL-17A and of cytotoxic molecules like granzyme B and perforin, as well as maintaining barrier integrity and participating in tissue repair and wound healing[16–20]. Recent transcriptome profiling discovered that TCR-mediated activation of MAIT cells leads to an upregulation of gene signatures associated with tissue repair[21–23], while cytokine-mediated activation enhances the expression of cytotoxic molecules in MAIT cells[14,24]. This implies that the mode of MAIT cell activation might determine their dichotomous function in different inflammatory contexts.

In people with MS, MAIT cells have been shown to be functionally altered in the blood, including increased production of IL-17A and upregulation of integrins and chemokine receptors involved in CNS migration[25]. MAIT cell frequencies in peripheral blood of people with MS were found to be decreased[26], but also increased[27] or unchanged[28]. These discrepancies are probably due to differences in the cohorts of patients studied. Although MAIT cells have been identified in MS brain lesions by different groups using different techniques[29–31], their functional characteristics and activation status in the CNS remain elusive. The role of MAIT cells in the EAE model has so far only been addressed by one study, in which *Mr1*−/− mice, devoid of MAIT cells due to the lack of positive selection in the thymus in the absence of antigen presentation, had an exacerbated disease course, implying a protective role of MAIT cells in EAE[32]. However, *Mr1*−/− mice show increased intestinal barrier permeability[18] and an altered microbiome composition[33] most likely resulting in additional immune cell dysregulations precluding any firm conclusions. Thus, conclusive phenotypic and functional analyses addressing the role of MAIT cells in EAE are still lacking.

Here, we use MR1-5-OP-RU tetramers to directly identify and characterize MAIT cells in EAE. CNS infiltrating MAIT cells are strongly activated via cytokines and TCR and possess inflammatory as well as tissue repair and protective features, while the latter outweigh the inflammatory effector potential in this disease model. The protective function of MAIT cells in EAE can even be boosted by 5-OP-RU treatment. Of note, CNS-infiltrating MAIT cells upregulate amphiregulin (AREG) upon TCR mediated activation in EAE. Together, this commends antigen-specific MAIT cell modulation as potential anti-inflammatory and tissue repair treatment strategy.

## Results
### Activated MAIT17 and MAIT1/17 cells accumulate in the inflamed CNS during EAE
First, we investigated whether MAIT cells are activated in EAE and contribute to the immune cell infiltrate in the CNS. We used 5-OP-RU-loaded MR1 tetramers[9] to specifically identify MAIT cells by flow cytometry of immune cells isolated from the CNS and lymph nodes (LN) in EAE and healthy C57BL/6 mice (Fig. 1a; Supplementary Fig. 1a). The gating strategy depicted in Supplementary Fig. 1a was used throughout the manuscript. Staining specificity was verified with cells from *Mr1*-deficient *(Mr1*−/−*)* mice and with MR1-6-FP tetramers as negative controls (Fig. 1a). In healthy mice, only very few T cells can be detected in the CNS, of which a small amount was identified as MAIT cells (0.26% ± 0.10% of αβ T cells, Fig. 1b, c). However, during the acute phase of EAE, MAIT cell frequencies among T cells in the CNS increased more than 13-fold (3.42% ± 1.84% of αβ T cells) and they were significantly enriched in the CNS compared to LN (Fig. 1b, c). We also detected MAIT cells in the meninges, but their frequency was unaltered in EAE (Supplementary Fig. 1b). Since MAIT cells leave the thymus as pre-primed cells with a memory phenotype (CD44+)[34], we always compared them to CD44+, activated non-MAIT T cells (gated as living CD45+CD11b−CD45R−TCR-β+MR1tetramer(5-OP-RU)−CD44+ cells, Supplementary Fig. 1a) in our model, termed non-MAIT T cells in the following. CNS-infiltrating MAIT cells were strongly activated as shown by the induction of CD69 expression, which was already evident at EAE onset, while in CNS-infiltrating non-MAIT T cells CD69 induction only became significant in the acute phase. In both cell types high CD69 expression in the CNS persisted in the chronic phase, possibly a sign of tissue residency. In the acute phase, CD69 expression by MAIT cells in the CNS was significantly higher than in non-MAIT T cells (Fig. 1d, e). Also, it increased more than 5-fold in comparison to LN, while there was only an approximately 3-fold increase in LN in EAE in comparison to healthy LN (Fig. 1f). As an additional indicator of activation, PD-1 expression was also strongly upregulated in CNS-infiltrating MAIT cells compared to MAIT cells from LN in acute EAE (Supplementary Fig. 1c, d). Next, we used adoptive transfer EAE without active immunization, allowing us to probe the role of *M. tuberculosis*-containing adjuvant for MAIT cell accumulation and activation in the inflamed CNS. Notably, MAIT cell frequency and CD69 expression were comparable in this model, rejecting the possibility that the *M. tuberculosis*-containing adjuvant is responsible for the CNS enrichment of MAIT cells (Supplementary Fig. 1e–g).

Murine MAIT cells can phenotypically be subdivided based on the expression of surface markers such as CD4, CD8, TCR-Vβ6 and TCR-Vβ8 or the transcription factors T-bet and RORγt. In healthy mice, the majority of MAIT cells belong to either T-bet+RORγt− MAIT1 or T-bet−RORγt+ MAIT17 cell subsets, which vary physiologically in their frequency in different tissues[16,34]. In our disease model, there was no preferential enrichment or expansion of certain MAIT cell subsets specified by surface markers (Supplementary Fig. 1h, i). By contrast, there was a strong tissue-specific increase of RORγt expression by MAIT cells in the CNS in general with almost all MAIT cells being RORγt+ in healthy and inflamed CNS (Supplementary Fig. 1j). Furthermore, part of MAIT cells in both healthy and inflamed CNS co-expressed RORγt and T-bet and can therefore be classified as MAIT1/17 cells (Fig. 1g, h). The frequency of CNS MAIT1/17 cells was significantly increased in acute EAE (Fig. 1h), and they expressed higher levels of CD69 than MAIT17 cells in the inflamed CNS (Fig. 1i). MAIT1/17 cells have previously been detected in other inflammatory models, such as bacterial infections[13,35]. Together with the description of CD4+RORγt+T-bet+ αβ T cells as pathogenic Th17 cells and main drivers of EAE progression[4,5], this could imply a similar inflammatory function of MAIT1/17 cells in EAE. However, the EAE disease course of MAIT cell-deficient *Mr1*−/− mice was aggravated during the acute phase in comparison to wildtype littermates (Supplementary Fig. 2a−c), as has also previously been described[32]. At the same time, the frequency and absolute numbers of CNS-infiltrating immune cell subsets were not significantly altered at EAE onset (Supplementary Fig. 2d, e) and during acute EAE in these mice (Supplementary Fig. 2f, g).

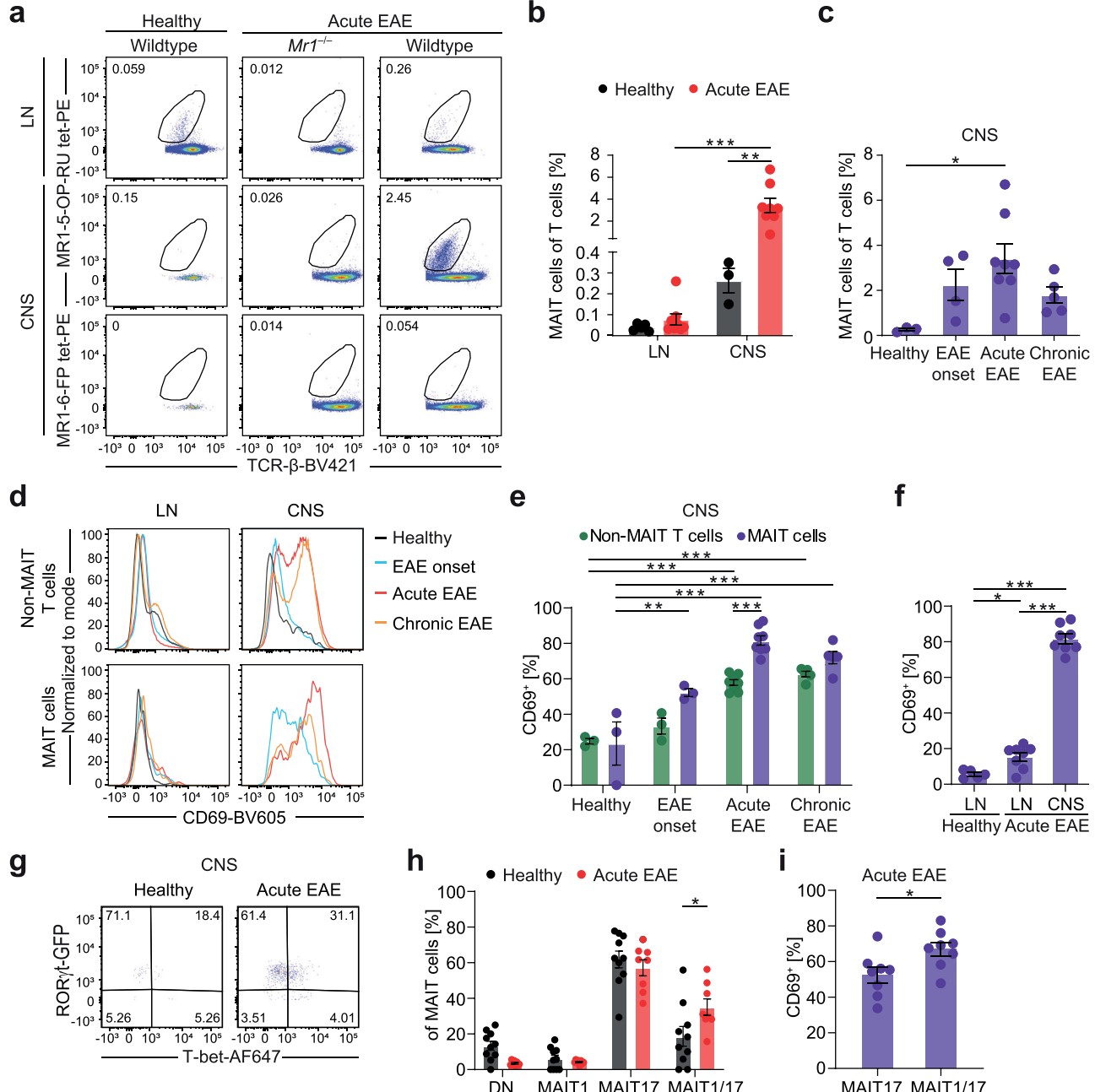

**Fig. 1 | Activated MAIT17 and MAIT1/17 cells accumulate in the inflamed CNS during EAE. a, b** MAIT cell frequency in LN and CNS of wildtype and $Mr1^{-/-}$ C57BL/6 J mice was analyzed by flow cytometry in healthy mice (LN, $n = 5$; CNS, $n = 3$ samples pooled from 3 mice per sample) and mice during acute EAE ($n = 8$; 14 days post immunization (dpi)). MAIT cells were gated as living CD45+CD11b−CD45R−TCR-β+MR1tetramer(5-OP-RU)+ cells. **c** MAIT cell frequency in the CNS of C57BL/6 mice was analyzed as described in (**a**) in healthy mice ($n = 3$ samples pooled from 3 mice per sample), mice at EAE onset ($n = 4$; 9 dpi), during acute EAE ($n = 8$; 14 dpi) and during chronic EAE ($n = 5$; 30 dpi). **d–f** T cell activation, reflected by CD69 expression, was quantified by flow cytometry in LN and CNS from healthy mice (LN, $n = 5$; CNS, $n = 3$), mice at EAE onset ($n = 3$), during acute EAE ($n = 8$) and during chronic EAE ($n = 5$). MAIT cells were gated as described in **a** and non-MAIT T cells were defined as living CD45+CD11b−CD45R−TCR-β+MR1tetramer(5-OP-RU)−CD44+ cells. **g, h** RORγt and T-bet expression were quantified in RORγtGFP transgenic reporter mice by flow cytometry after intra-nuclear T-bet staining in CNS of healthy mice ($n = 10$) and of mice during acute EAE ($n = 8$). MAIT cells were classified as double-negative (DN; RORγt−T-bet−), MAIT1 (RORγt−T-bet+), MAIT17 (RORγt+T-bet−) and MAIT1/17 (RORγt+T-bet+). **i** The frequency of CD69+ cells was quantified among MAIT17 and MAIT1/17 cells in the CNS of RORγtGFP transgenic reporter mice during acute EAE ($n = 8$). Data are shown as mean ± SEM. Statistics: two-way ANOVA in (**b, e, h**) (**h**, $P = 0.0116$); one-way ANOVA in **c** ($P = 0.0278$), **f**; t-test (two-tailed) in **i** ($P = 0.0243$); *$P < 0.05$, **$P < 0.01$, ***$P < 0.001$. Source data are provided as a Source Data file.

In summary, activated MAIT cells accumulate in the CNS in acute EAE and belong to the MAIT17 and MAIT1/17 subsets. The detrimental effect of their absence in EAE implies a protective function in sterile CNS inflammation.

## Inflammatory and tissue repair phenotype of CNS-infiltrating MAIT cells in EAE

To disentangle the discrepancy of inflammatory phenotypic features, but a potentially protective role of MAIT cells in EAE, we next set out

to more broadly address their functional phenotype. For this purpose, we performed bulk RNA-sequencing of MAIT cells isolated from the CNS and spleen during the acute phase of the disease as well as from the spleen of healthy mice. For this experiment, the spleen was chosen as peripheral lymphoid organ instead of LN to reach sufficiently high numbers of MAIT cells for sequencing. Principal component analysis (PCA) revealed a high consistency within the groups (Supplementary Fig. 3a), whereas several genes were differentially expressed between the groups (Supplementary Fig. 3b). Especially MAIT cells that were isolated from the CNS could be readily distinguished from MAIT cells from the spleen. This suggests an induction of a specific transcriptional profile in MAIT cells infiltrating the inflamed CNS. This transcriptional profile included genes associated with both, inflammatory as well as tissue repair function,

as we found an enrichment of respective GO terms comparing MAIT cells from the inflamed CNS with those from the spleen of EAE animals (Fig. 2a).

A potential dual role of MAIT cells infiltrating the inflamed CNS was further corroborated by gene set enrichment (GSEAs) and AUCell analyses. In line with the MAIT1/17 phenotype described above, GSEA revealed that a previously described gene set of pathogenic Th17 cells[4] was enriched in MAIT cells isolated from the spleen of EAE mice in comparison to MAIT cells from the healthy spleen. This enrichment was also observed in MAIT cells from the CNS in EAE compared to those from the healthy or EAE spleen (Fig. 2b). At the same time, a gene set described to be specifically associated with tissue repair function of commensal-specific CD8[+] T cells[36] as well as a third gene set, more generally reflecting tissue repair[37], were equally increased in MAIT cells

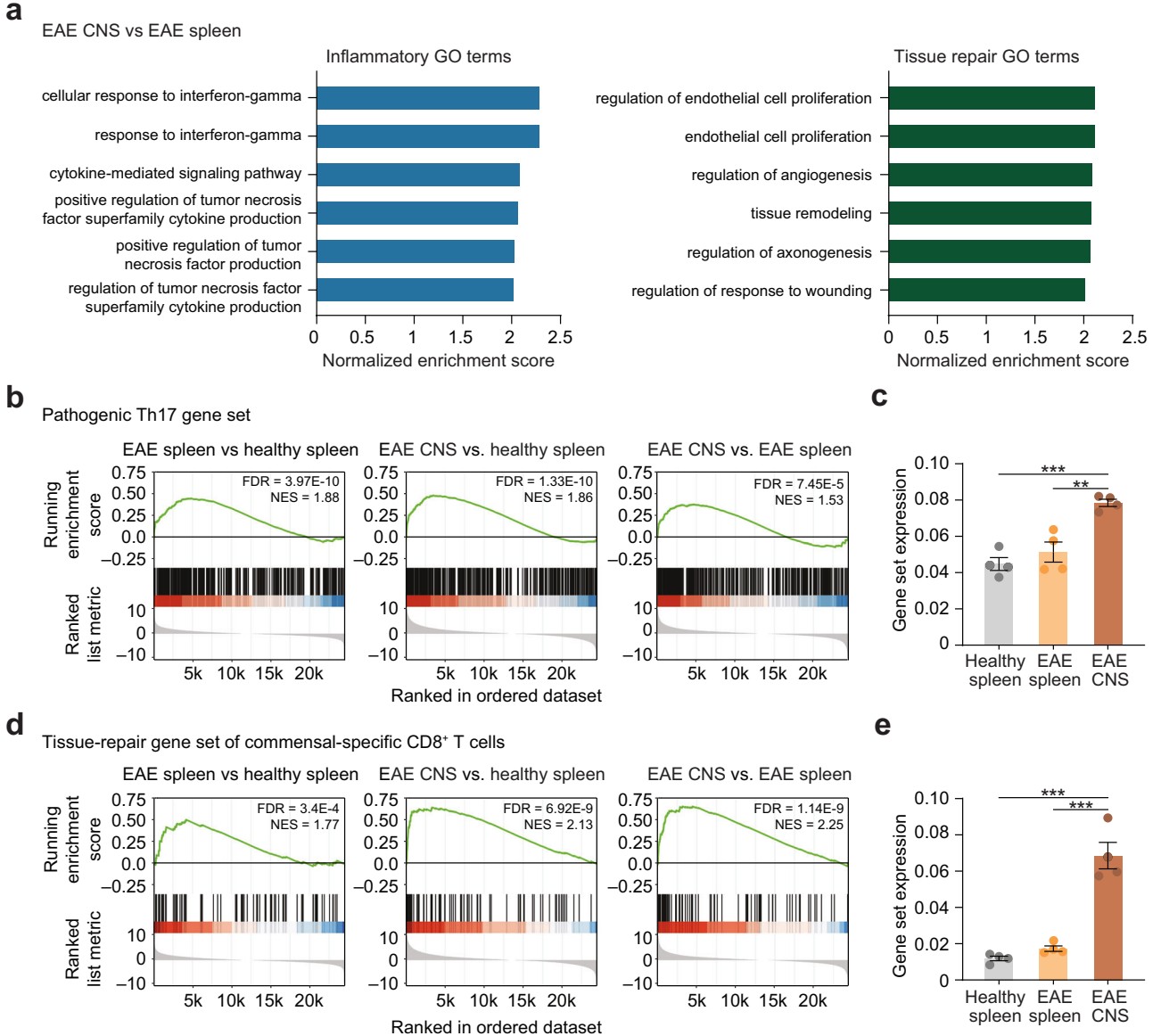

**Fig. 2 | Induction of inflammatory and tissue repair gene signatures in CNS-infiltrating MAIT cells in EAE.** MAIT cells from healthy spleen, EAE spleen and EAE CNS during acute EAE (14 days post immunization) of C57BL/6 J mice ($n = 4$ samples per group pooled from 5 mice per sample) were sorted (living CD45[+]CD11b[−]CD45R[−]TCR-β[+]MR1tetramer(5-OP-RU)[+] cells) and subjected to bulk-RNA sequencing. **a** Inflammatory and tissue repair associated GO-terms among the 50 most upregulated GO terms represented in differentially expressed genes between EAE CNS and EAE spleen. **b**, **d** Gene set enrichment analysis (GSEA) of

signatures defining pathogenic Th17 cells[3] and tissue repair[33] in indicated comparisons. Normalized enrichment score (NES) and false discovery rate (FDR; P-values after Benjamini-Hochberg adjustment) are shown. **c**, **e** Quantification of expression of indicated gene sets in MAIT cells derived from healthy spleen, EAE spleen and EAE CNS by AUCell analysis of data generated as described above. Data are shown as mean ± SEM. Statistics: gene set enrichment analysis (GSEA) in (**b**, **d**); one-way ANOVA in (**c**, **e**) **P < 0.01, ***P < 0.001. Source data are provided as a Source Data file.

from the spleen and CNS in EAE (Fig. 2d and Supplementary Fig. 3c). To increase robustness of our analyses and to be able to compare the expression of gene sets in all three conditions at the same time, we further performed AUCell analyses. A significant upregulation of all three gene sets only in MAIT cells from the CNS in EAE compared to those from the spleen of both EAE and healthy mice was confirmed by AUCell, while there was no significant difference in this analysis between MAIT cells from the spleens of EAE and healthy mice (Fig. 2c, e and Supplementary Fig. 3d).

To validate the pro-inflammatory and tissue repair transcriptional profiles of CNS-infiltrating MAIT cells on protein level, we selected candidate effector molecules reflecting the respective functions. Cytokine production is one of the major effector functions of T cells, and the cytokines GM-CSF, IL-17A and IFN-γ have previously been described as inflammatory in EAE[38]. IL-22 has been associated with pathogenic Th17 cells[39], but seems to have a protective effect in EAE[40]. Our transcriptome data showed that MAIT cells from the CNS expressed significantly higher levels of *Il22*, *Csf2* and *Il17a* in comparison to MAIT cells from the spleen in EAE, whereas *Ifng* was unaltered (Fig. 3a). We could confirm this enhanced type-17 like phenotype of MAIT cells from the inflamed CNS by intracellular cytokine staining after PMA and ionomycin stimulation. They produced significantly

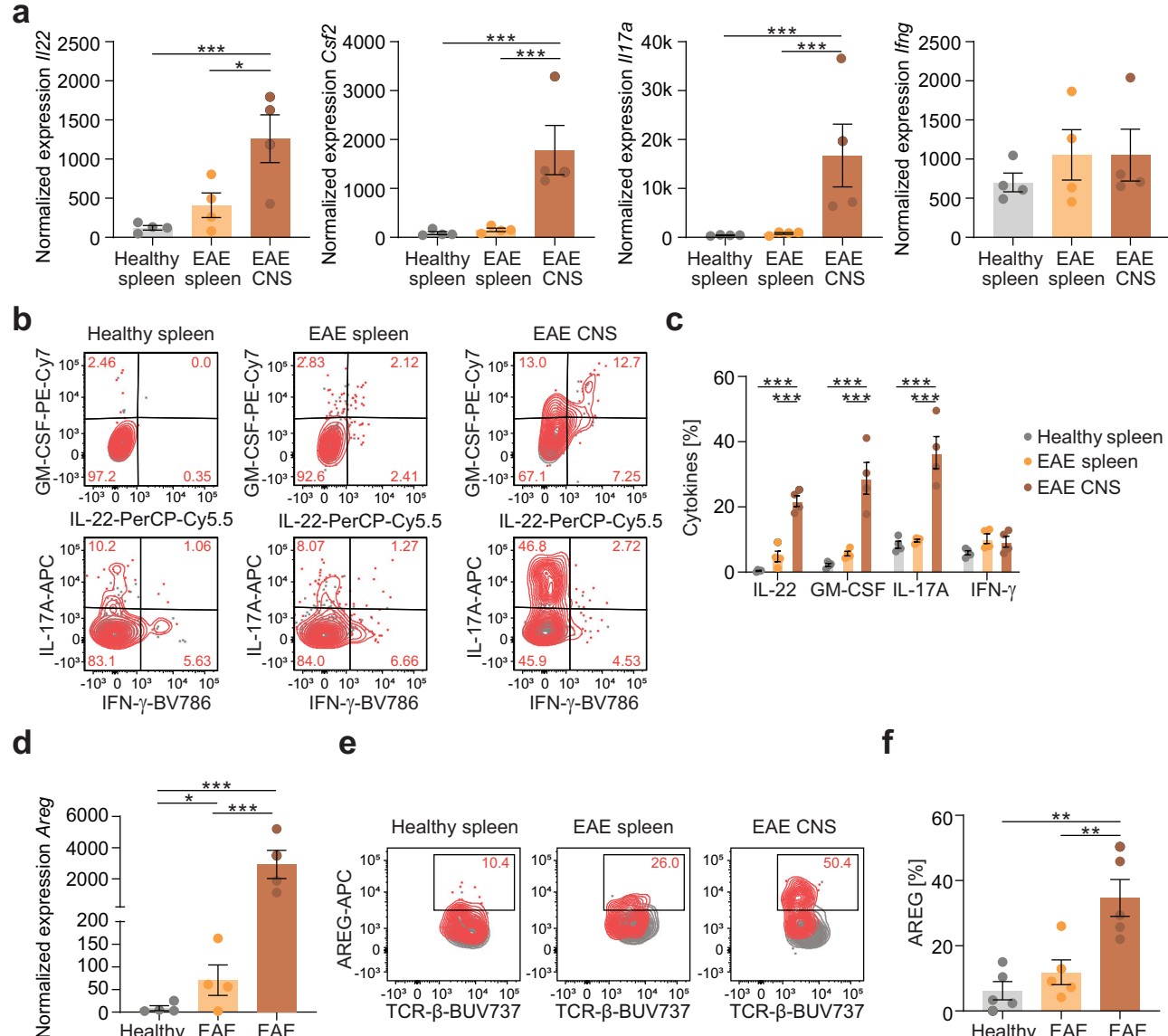

**Fig. 3 | Induction of effector molecules representing inflammatory and tissue repair function by MAIT cells in acute EAE. a** Normalized expression of *Il22*, *Csf2*, *Il17a* and *Ifng* from RNA sequencing data of MAIT cells sorted from healthy spleen, EAE spleen and EAE CNS of C57BL/6 J mice (n = 4 samples per group pooled from 5 mice per sample). **b, c** MAIT cells were isolated from healthy spleen and from spleen and CNS during acute EAE of C57BL/6 J mice (n = 4 per group) and were cultured for 4 h in the presence of phorbol 12-myristate-13-acetate (PMA) (10 ng/ml), ionomycin (1 μg/ml) and monensin (2 μM) (red) or left unstimulated (grey). IL-22, GM-CSF, IL-17A and IFN-γ were subsequently stained intracellularly and quantified by flow cytometry. **d** Normalized expression of *Amphiregulin* (*Areg*) from RNA sequencing data of sorted MAIT cells from C57BL/6 J mice (n = 4 samples per group pooled from 5 mice per sample). **e, f** MAIT cells were isolated from healthy spleen and from spleen and CNS during acute EAE of C57BL/6 J mice (n = 5 per group) and were cultured for 4 h in the presence of PMA (10 ng/ml), ionomycin (1 μg/ml) and monensin (2 μM) (red) or left unstimulated (grey). AREG was subsequently stained intracellularly and quantified by flow cytometry. Data are shown as mean ± SEM. Statistics: DESeq2 false discovery rate-adjusted *P*-value in (**a, d**); two-way ANOVA in **c**; one-way ANOVA in (**f**); *P < 0.05, **P < 0.01, ***P < 0.001. Source data are provided as a Source Data file.

higher levels of IL-22, GM-CSF and IL-17A compared to MAIT cells from the spleen of healthy mice and of mice in acute EAE, while IFN-γ production was unchanged (Fig. 3b, c). Notably, MAIT cells from the inflamed CNS produced even higher levels of IL-17A and GM-CSF than other CNS-infiltrating T cells, as well as IL-22 at comparable levels, while IFN-γ was decreased in comparison to non-MAIT T cells (Supplementary Fig. 3e, f). As an effector molecule reflecting the tissue repair transcriptional profile of MAIT cells in the CNS in EAE, we selected *Amphiregulin* (*Areg*) from the top ten leading edge genes of our GSEAs (Supplementary Fig. 3g). AREG production by Tregs has previously been shown to mediate neuronal recovery and therefore contribute to regeneration in an ischemic stroke model[41]. Furthermore, microglia-derived AREG has been identified as suppressor of pathogenic astrocyte responses in EAE[42]. In our transcriptome data, *Areg* expression was significantly increased in MAIT cells from the inflamed CNS, when plotted at single gene level (Fig. 3d). Using flow cytometry, we confirmed a significant induction of AREG on protein level in MAIT cells isolated from the inflamed CNS, as compared to those isolated from the spleen of healthy mice or from the spleen of mice during acute EAE (Fig. 3e, f).

Taken together, our transcriptome data and selective hit validation on protein level by flow cytometry corroborate that MAIT cells phenotypically possess inflammatory as well as tissue repair and protective features in the inflamed CNS.

## TCR-mediated activation of MAIT cells in EAE

The induction of tissue repair gene signatures in MAIT cells has recently been associated specifically with TCR activation in contrast to cytokine-mediated activation[21–23,43]. To investigate whether MAIT cells in acute EAE get activated via their TCR, we first compared our transcriptome data with the published transcriptome datasets of differentially activated human MAIT cells[22]. Indeed, GSEA and AUCell analyses indicate that MAIT cells in EAE were activated via their TCR in combination with cytokine-mediated activation. All three gene signatures were significantly increased in MAIT cells isolated from the inflamed CNS in comparison to MAIT cells from the healthy and EAE spleen (Fig. 4a; Supplementary Fig. 4a). The gene signature reflecting TCR activation alone was only significantly upregulated in CNS-infiltrating MAIT cells, while the other two signatures were also significantly increased in MAIT cells from the spleen in EAE in comparison to those from the healthy spleen (Fig. 4a). Given the large number of overlapping genes in the TCR- and cytokine-activated MAIT cell signatures[22], we extracted gene signatures exclusively present in either the TCR activation or cytokine activation signature by subtracting overlapping genes. GSEA and AUCell analyses showed that also these signatures were both enriched in CNS-infiltrating MAIT cells in EAE, and that only the cytokine driven signature was also enriched in MAIT cells from the spleen in EAE (Supplementary Fig. 4b, c, d).

TCR-mediated activation of T cells exclusively leads to the induction of *Nr4a1* (nuclear hormone receptor subfamily 4 group A member 1, Nur77)[44]. Notably, *Nr4a1* was increased in MAIT cells from the inflamed CNS compared to MAIT cells from the healthy spleen in our transcriptome dataset (Supplementary Fig. 4e). To more closely investigate TCR-mediated activation of MAIT cells in EAE, we used Nur77GFP reporter mice, which transiently express GFP after TCR activation, but not in response to other inflammatory stimuli like lipopolysaccharide (LPS) and cytokines[45]. In vitro stimulation of liver-derived cells from these mice with the MAIT-specific TCR antigen 5-OP-RU indeed induced a concentration-dependent increase of Nur77GFP in MAIT but not in non-MAIT T cells (Supplementary Fig. 4f). Comparison of TCR-mediated activation of MAIT cells with activation in a cytokine-dependent manner using IL-12 and IL-18 or with a combination of both revealed a significant upregulation of Nur77GFP only in response to TCR activation (Supplementary Fig. 4g). Furthermore, when followed by PMA and ionomycin stimulation, TCR-mediated and cytokine-dependent activation both lead to significantly more IL-17A and AREG expression by MAIT cells in vitro, whereas GM-CSF was only increased after cytokine-dependent activation. For AREG we observed an additive effect of cytokine-dependent activation combined with TCR-mediated activation (Supplementary Fig. 4h).

In EAE, we could detect a significant increase of Nur77GFP-expressing MAIT cells in the CNS already in the preclinical phase (6–7 days post immunization) of EAE, while the peak of Nur77-induction in non-MAIT T cells in the CNS was reached in the acute phase (13–14 days post immunization). In both, MAIT and non-MAIT T cells, Nur77 expression in the CNS declined in the chronic phase of the disease (30 and 45 days post immunization) (Fig. 4b, c). In LN, we found an induction of Nur77 in non-MAIT T cells in the preclinical phase, which lasted during the acute phase and declined in chronic EAE. By contrast, there was no detectable induction in MAIT cells from LN (Supplementary Fig. 4i). In comparison to LN and spleen, there was a more than 3- and 4-fold increase in Nur77-expression in MAIT cells from the CNS in EAE, respectively. Of note, other innate-like T cells, namely γδ-T cells and NKT cells, did not show the same extent of Nur77 expression in the inflamed CNS, while the expression by MAIT cells was comparable to activated classical αβ T cells, which include disease-initiating autoreactive T cells (Fig. 4d). MAIT cells isolated from dural meninges during acute EAE expressed significantly lower levels of Nur77 compared to CNS MAIT cells (Supplementary Fig. 4j, k).

To analyze whether the effector phenotype of MAIT cells in EAE is influenced by their mode of activation, we next sorted Nur77GFP− and Nur77GFP+ MAIT cells from the CNS in acute EAE and stimulated the cells with PMA and ionomycin. Nur77GFP-expressing MAIT cells produced significantly less IL-17A and more AREG than Nur77GFP− MAIT cells, whereas GM-CSF was not differentially expressed (Fig. 4e, f). Therefore, TCR-mediated MAIT cell activation in EAE seems to correlate with increased expression of AREG and decreased expression of IL-17A.

Together, we could confirm TCR-mediated activation of MAIT cells in the inflamed CNS evident from our transcriptome data by use of a specific reporter mouse in vivo. Furthermore, this activation might enhance the tissue repair potential and dampen the pro-inflammatory potential of MAIT cells in EAE.

## TCR activation enhances the protective effect of MAIT cells in EAE

Next, we set out to specifically interfere with this TCR-mediated activation of MAIT cells in vivo during EAE to investigate whether this mode of activation indeed drives MAIT cells to exert their protective and tissue repair potential in CNS inflammation. We first treated wildtype C57BL/6 J mice with i.p. injections of an anti-MR1 blocking antibody to specifically inhibit TCR-mediated activation of MAIT cells. We chose to treat at 5, 10 and 15 days post immunization to cover the timepoints with the strongest TCR-mediated activation of MAIT cells in EAE based on Nur77GFP reporter expression (see above). Indeed, MR1 blockade led to an exacerbated EAE course in the chronic phase in comparison to IgG isotype-treated littermates (Fig. 5a–c). To prove the specificity of this treatment in our model, we immunized and treated Nur77GFP reporter mice the same way. In these mice, Nur77 expression was specifically reduced after anti-MR1 treatment compared to IgG isotype control treatment in MAIT cells from the CNS in acute EAE, while there was no alteration of Nur77GFP expression in CNS-derived non-MAIT T cells (Fig. 5d, e).

Second, we aimed to specifically activate MAIT cells in a TCR-dependent manner to enhance their protective and tissue repair potential in a therapeutic approach by administration of 5-OP-RU. To first establish the specificity of this treatment in our animal model, we performed in vivo tests using Nur77GFP reporter mice. After application of 5-OP-RU, Nur77GFP expression was enhanced in MAIT cells in LN, spleen and CNS, but not in non-MAIT classical or innate-like T cells (Fig. 6a). In line with the exacerbation after MR1-blockade, 5-OP-RU

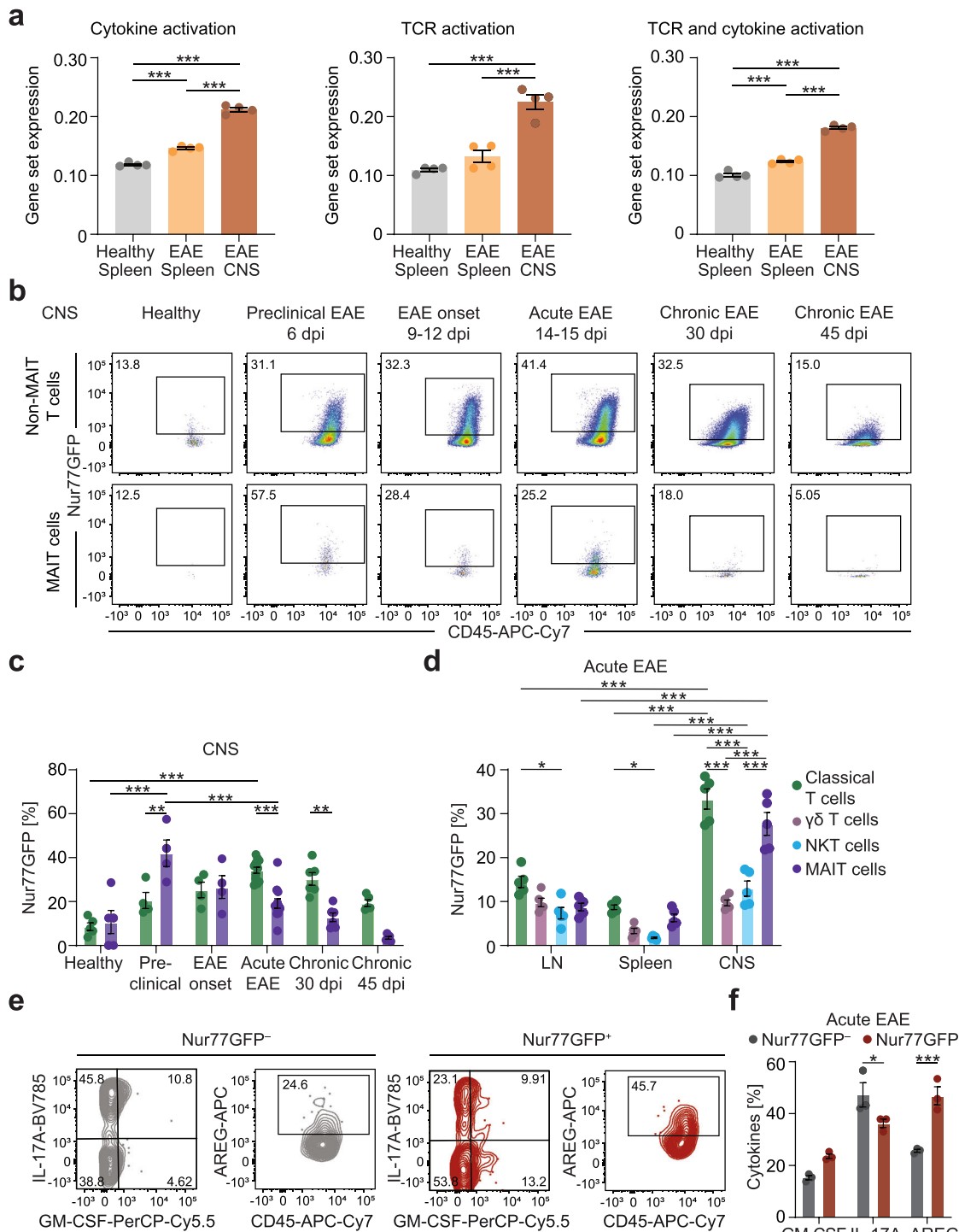

**Fig. 4 | TCR- and cytokine-mediated activation of MAIT cells in EAE. a** Expression of gene sets enriched in MAIT cells activated via cytokines, TCR or via cytokines and TCR[19] in sorted MAIT cells from indicated tissues of C57BL/6 J mice ($n = 4$ samples per group pooled from 5 mice per sample) quantified by AUCell analysis. **b**, **c** Nur77 expression in the CNS quantified by flow cytometry in healthy Nur77GFP reporter mice ($n = 5$ samples pooled from 3 mice per sample) and during preclinical EAE (6–7 days post immunization (dpi); $n = 4$), EAE onset (9–11 dpi; $n = 4$), acute EAE (13–14 dpi; $n = 12$) and chronic EAE (30 dpi, $n = 6$; 45 dpi, $n = 5$). **d** Nur77GFP expression was measured by flow cytometry in classical T cells (living CD45+CD11b−CD45R−TCR-β+MR1tetramer(5-OP-RU)−CD1dtetramer(PBS-57)−CD44+ cells), γδ T cells (living CD45+CD11b−CD45R−TCR-β−TCR-γδ+ cells), NKT cells (living CD45+CD11b−CD45R−TCR-β+CD1dtetramer(PBS-57)+ cells) and MAIT cells (living CD45+CD11b−CD45R−TCR-β+MR1tetramer(5-OP-RU)+ cells) from LN, spleen and CNS of Nur77GFP reporter mice during acute EAE (14 dpi; $n = 5$). **e**, **f** Nur77GFP− and Nur77GFP+ MAIT cells were sorted from the CNS of Nur77GFP reporter mice in acute EAE ($n = 3$ samples per cell type pooled from 4 mice per sample) and cultured for 4 h in the presence of phorbol 12-myristate-13-acetate (PMA) (10 ng/ml), iono-mycin (1 μg/ml) and monensin (2 μM). GM-CSF, IL-17A and AREG were subsequently stained intracellularly and quantified by flow cytometry. Data are shown as mean ± SEM. Statistics: one-way ANOVA in (**a**); two-way ANOVA in (**c**, **d**, **f**); *$P < 0.05$, **$P < 0.01$, ***$P < 0.001$. Source data are provided as a Source Data file.

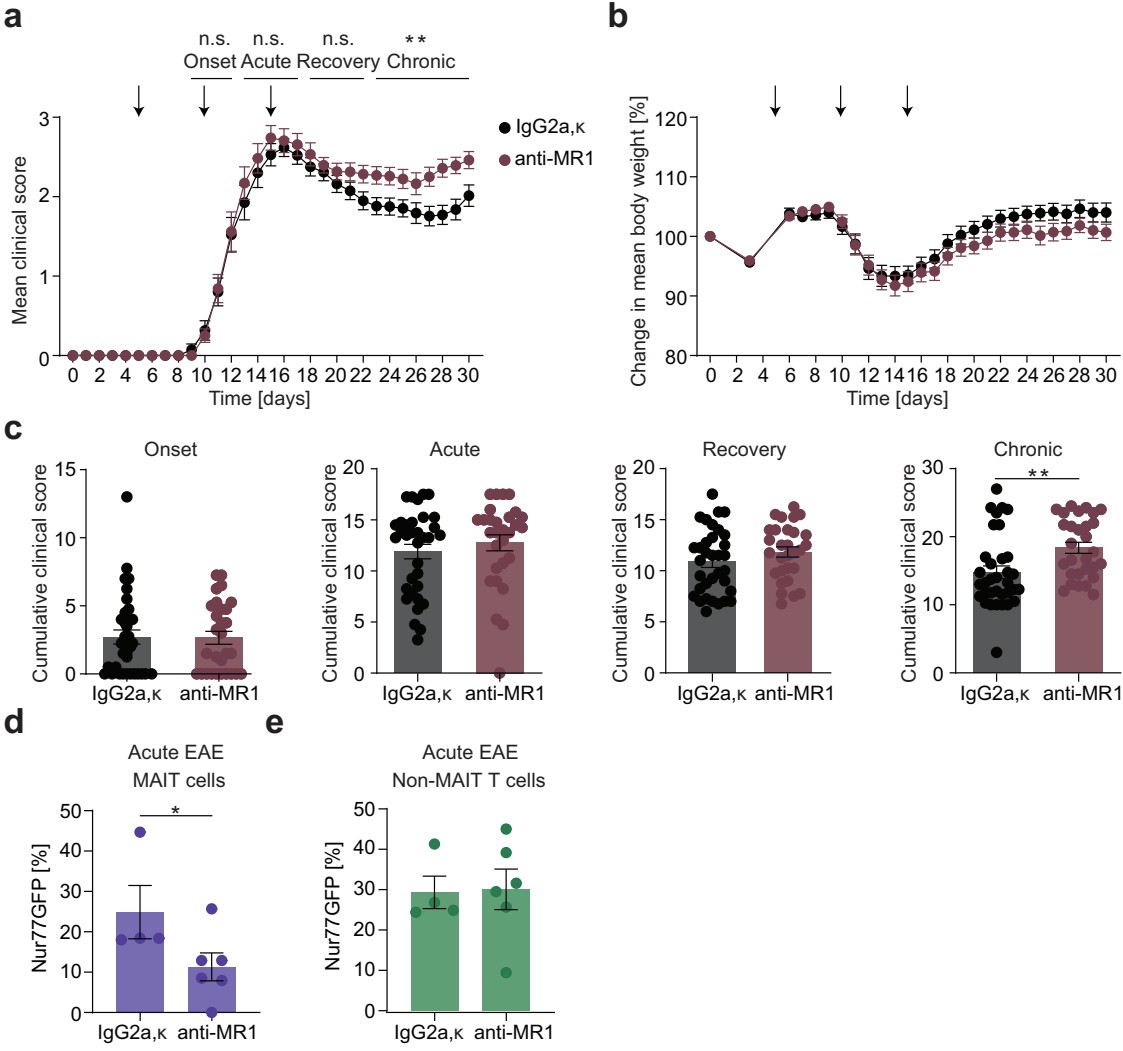

**Fig. 5 | Blocking TCR activation of MAIT cells exacerbates EAE. a–c** EAE was induced in female C57BL/6 J mice by active immunization against MOG$_{35\text{-}55}$ peptide. Clinical score and body weight were assessed daily. At 5, 10 and 15 days post immunization (dpi) mice were injected i.p. either with an anti-MR1 blocking antibody (*n* = 29) (clone: 26.5, 250 μg/injection) or with a respective IgG isotype control (*n* = 34). Data were pooled from two independent experiments. The EAE course was divided into EAE onset (day of first symptoms–12 dpi), acute EAE (13–17 dpi), EAE recovery (18–22 dpi) and chronic EAE (23–30 dpi). The cumulative score in the respective phases was compared between groups. **d, e** Nur77GFP reporter mice were immunized and treated with anti-MR1 blocking antibody (*n* = 6) or isotype control (*n* = 4) as described in (**a**). During acute EAE (16 dpi), Nur77GFP expression of MAIT and non-MAIT T cells from the CNS was analyzed. Data are shown as mean ± SEM. Statistics: Mann-Whitney-U test (two-tailed) in (**a, c**) (*P* = 0.0034); t-test (one-tailed) in (**d**) (*P* = 0.0404), **e**; *\*P* < 0.05, \*\**P* < 0.01. Source data are provided as a Source Data file.

treatment (1 nmol / injection) of EAE mice in a therapeutic approach beginning at disease onset (9 days post immunization) led to an ameliorated disease course in the chronic phase (Fig. 6b–d). While the effect on clinical symptoms was small at such low dose treatment, immunohistopathological analyses corroborated the therapeutic potential of 5-OP-RU. The area of glial fibrillary acidic protein (GFAP) was significantly reduced in cervical spinal cord sections 30 days post immunization in 5-OP-RU treated animals, representing a decrease in reactive astrogliosis (Fig. 6e, f).

As the effect of low dose 5-OP-RU treatment on the disease course was only moderate, we tested a treatment regimen using a higher dose (30 nmol / injection) for 5 consecutive days starting at EAE onset (9–13 days post immunization) (Fig. 7a, b). We chose to treat only during the phase of acute inflammation and for a short time period to avoid MAIT cell depletion at this higher dose. For these experiments we used the prodrug 5-A-RU-PABC-Val-Cit-Fmoc, which equally induces specific MAIT cell activation in vivo[46]. Indeed, this treatment regimen led to a stronger and immediate disease amelioration, which was already

evident in the acute and recovery phase lasting through the chronic phase (Fig. 7a–c). The frequency of MAIT cells was unaltered in treated animals excluding MAIT cell depletion (Fig. 7d). Linking the ameliorated disease course to MAIT cell tissue repair and protective function, AREG expression was elevated in CNS-infiltrating MAIT cells from 5-A-RU-PABC-Val-Cit-Fmoc treated mice compared to PBS treated mice, in contrast to non-MAIT T cells and microglia showing unaltered AREG expression (Fig. 7e). Moreover, we detected higher AREG levels in spinal cord tissue lysates of 5-A-RU-PABC-Val-Cit-Fmoc treated EAE mice (Fig. 7f), showing that increased AREG production by MAIT cells after TCR-mediated activation leads to increased total amounts of AREG in the target tissue of our model.

To summarize, blocking as well as activating MAIT cells via their TCR in EAE revealed that their tissue repair and protective function outweighed their pathogenic potential. This is likely mediated by AREG as an established suppressor of astrocyte pathogenic function in EAE and can potentially be targeted therapeutically by enhancing their cognate antigen activation.

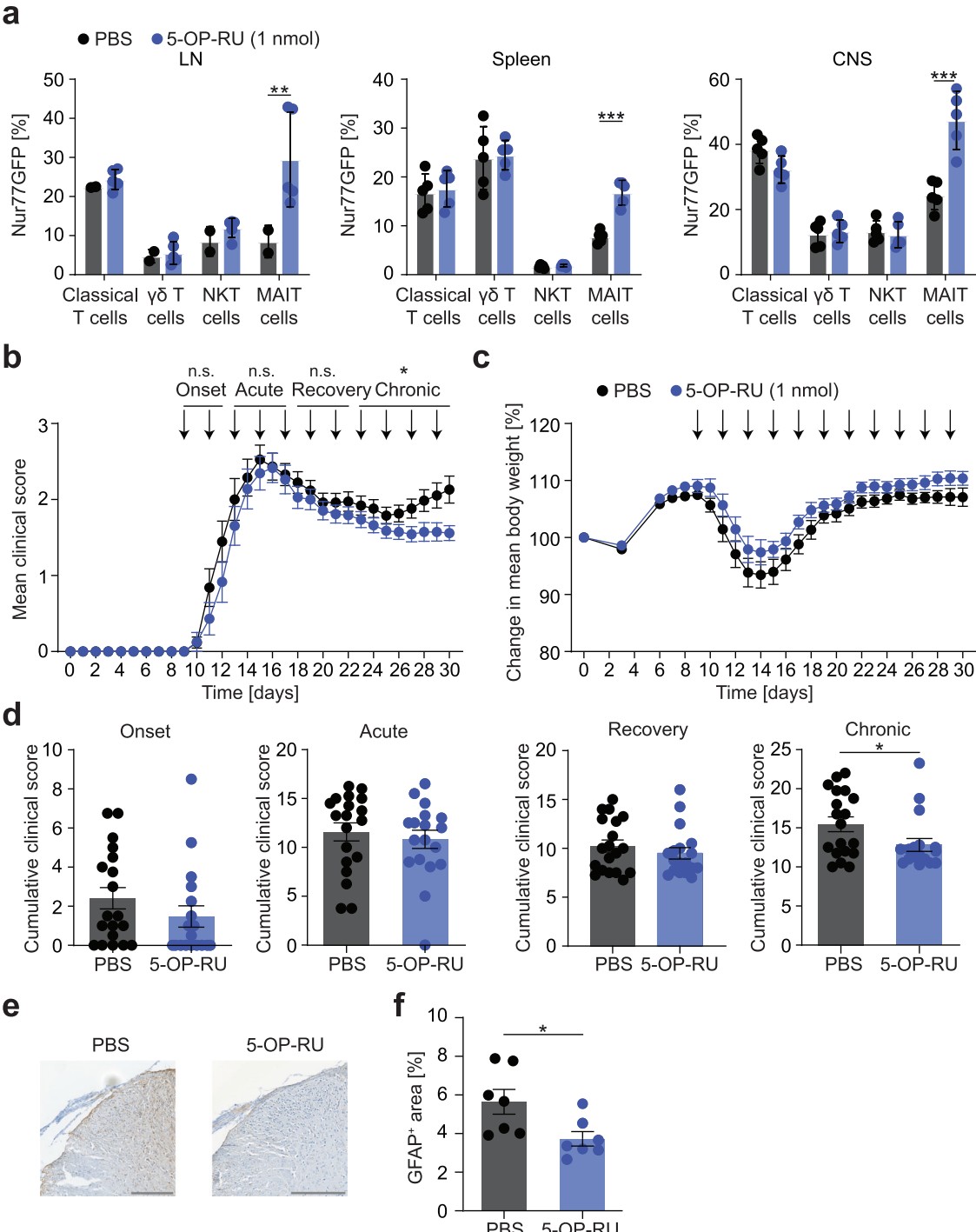

**Fig. 6 | TCR activation of MAIT cells ameliorates chronic EAE. a** EAE was induced in Nur77GFP reporter mice by active immunization against MOG$_{35\text{-}55}$ peptide, followed by i.p. injections of 5-OP-RU (1 nmol) or PBS every 2 days starting at EAE onset (9 days post immunization (dpi)). During acute EAE (15 dpi), the Nur77GFP signal of classical T cells (living CD45$^+$CD11b$^-$CD45R$^-$TCR-β$^+$MR1tetramer(5-OP-RU)$^-$CD1dtetramer(PBS-57)$^-$CD44$^+$ cells), γδ T cells (living CD45$^+$CD11b$^-$CD45R$^-$TCR-β$^-$TCR-γδ$^+$CD44$^+$ cells), NKT cells (living CD45$^+$CD11b$^-$CD45R$^-$TCR-β$^+$CD1dtetramer(PBS-57)$^+$CD44$^+$ cells) and MAIT cells was measured by flow cytometry in LN (PBS, $n$ = 2 mice; 5-OP-RU, $n$ = 5 mice), spleen ($n$ = 5 mice per group) and CNS ($n$ = 5 mice per group). **b–d** EAE was induced in female C57BL/6 J mice by active immunization against MOG$_{35\text{-}55}$ peptide. Mice were injected i.p. with 5-OP-RU (1 nmol, $n$ = 18) or PBS ($n$ = 19) every 2 days starting at EAE onset (9 dpi). The cumulative score of the respective phases divided as in Fig. 5 was compared between groups. **e, f** 30 dpi, cervical spinal cord slices of female C57BL/6 J mice ($n$ = 7 per group) were stained for GFAP. Scale bar, 250 μm. Area of GFAP was quantified using ImageJ. Data are shown as mean ± SEM. Statistics: two-way ANOVA in (**a**) (LN, $P$ = 0.0016; Spleen, $P$ = 0.0006; CNS, $P$ < 0.00001); Mann-Whitney-U test (two-tailed) in (**b**, **d**) ($P$ = 0.0478), (**f**) ($P$ = 0.0175); *$P$ < 0.05, **$P$ < 0.01, ***$P$ < 0.001. Source data are provided as a Source Data file.

## Discussion

Here we show that MAIT cells are activated in EAE and infiltrate the CNS with a phenotypical inflammatory as well as tissue repair and protective potential. Functionally, protective effects of MAIT cells overrule their pathogenic potential in vivo and this protective function is at least in part driven by cognate TCR activation and might be mediated by AREG.

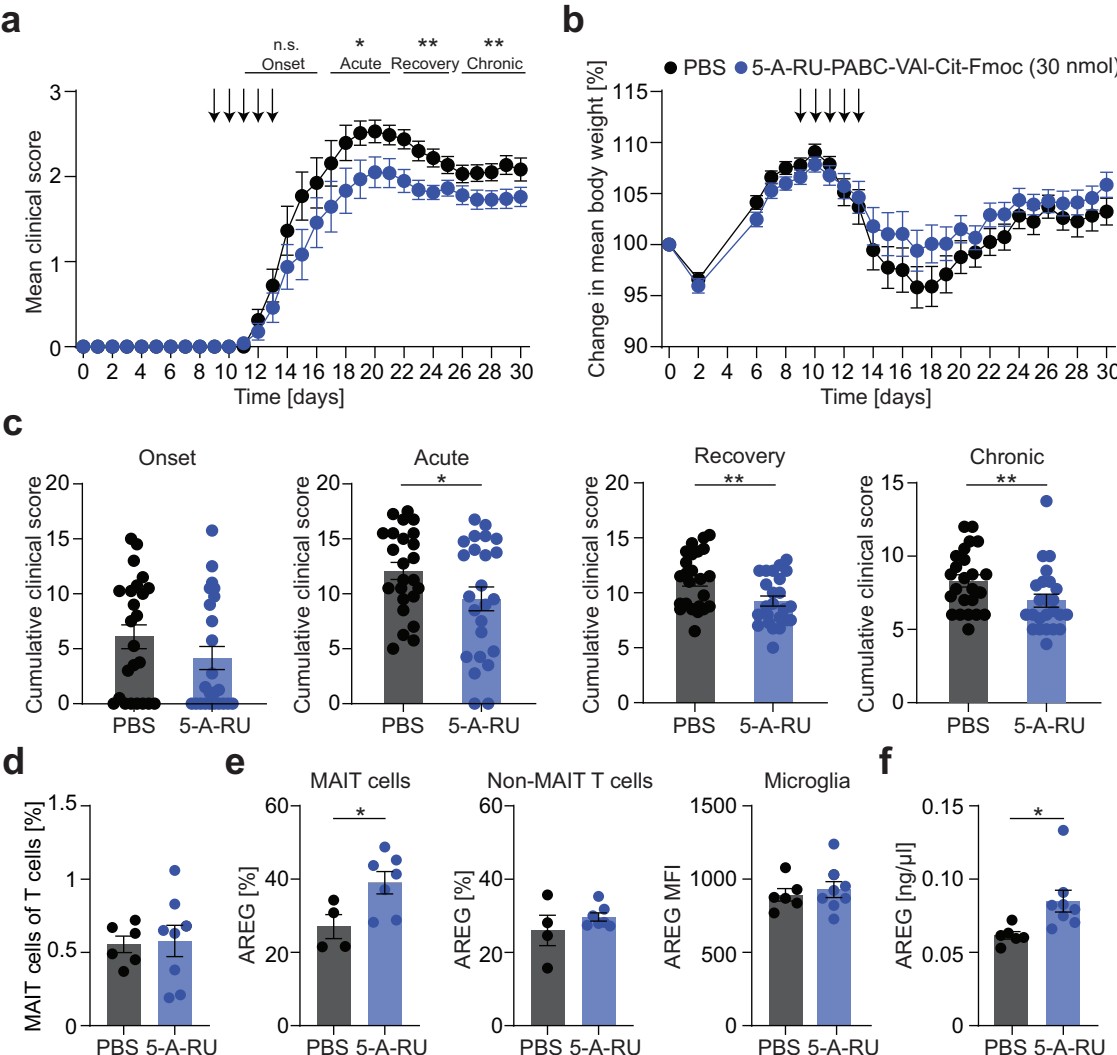

**Fig. 7 | Protective effect of high-dose TCR-activation in EAE. a–c** EAE was induced in female C57BL/6 J mice by active immunization against $MOG_{35-55}$ peptide. Mice were injected i.p. with 5-A-RU-PABC-Val-Cit-Fmoc (30 nmol, $n = 25$) or PBS ($n = 25$) for 5 days starting at EAE onset (9 days post immunization (dpi)). The EAE course was divided into EAE onset (day of first symptoms–16 dpi), acute EAE (17–21 dpi), EAE recovery (22–25 dpi) and chronic EAE (26–30 dpi). The cumulative score of the respective phases was compared between groups. **d–f** EAE was induced in female C57BL/6 J mice and the mice were treated as described in **a** and analyzed in acute EAE (20 dpi). **d** MAIT cell frequency in the CNS of PBS ($n = 6$) or 5-A-RU-PABC-Val-Cit-Fmoc ($n = 8$) treated mice was analyzed by flow cytometry. **e** Cells were isolated from the CNS of PBS ($n = 4$) or 5-A-RU-PABC-Val-Cit-Fmoc (30 nmol; $n = 7$) treated mice and cultured for 4 h in the presence of phorbol 12-myristate-13-acetate (PMA) (10 ng/ml), ionomycin (1 μg/ml) and monensin (2 μM). Intracellular AREG of MAIT cells, non-MAIT T cells and microglia was measured by flow cytometry. **f** Amounts of AREG in spinal cord lysates of PBS ($n = 6$) or 5-A-RU-PABC-Val-Cit-Fmoc ($n = 8$) treated mice were quantified by ELISA. Data are shown as mean ± SEM. Statistics: Mann-Whitney-U test (one-tailed) in (**a**, **c**) (Acute, $P = 0.0393$; Recovery, $P = 0.0066$; Chronic, $P = 0.0077$); t-test (two-tailed) in (**d**, **e**) (**e**, $P = 0.0323$), (**f**) ($P = 0.0224$); *$P < 0.05$, **$P < 0.01$. Source data are provided as a Source Data file.

While MAIT cells were present only at very low frequencies in the CNS of healthy mice, they accumulated to about 3.42% ± 1.84% of αβ T cells in the inflamed CNS in EAE representing a 13-fold increase. In bacterial infection models of the lung and skin, where MAIT cells are present at higher frequencies already in the healthy state, they can expand to up to 25–50% of all T cells representing a 30-fold and 4-fold increase, respectively[13,19,35]. However, MAIT cell accumulation also occurs in other disease models where they are nearly absent in healthy tissues. For example, MAIT cells infiltrate the retina in an experimental autoimmune uveitis (EAU) model (~1.5% of T cells) where they equally seem to exert a protective function, which is at least partially mediated by IL-22 production[47]. Therefore, further investigations into the role of MAIT cells in organ-specific autoimmune or inflammatory diseases with a low MAIT cell frequency in the healthy tissue are warranted.

Our further phenotypic and functional characterization of CNS-infiltrating MAIT cells in EAE revealed their strong activation mediated by a combination of cytokine and TCR signals. The cytokines IL-12 and IL-18 can directly and TCR-independently activate MAIT cells[11], and both of these cytokines are elevated in EAE as well as in people with MS[48,49]. By using Nur77GFP reporter mice, we could show that about 30% of CNS-infiltrating MAIT cells were activated via their TCR in EAE. This TCR-mediated activation was not a general phenomenon of all T cells in the inflamed CNS, as we did not detect it in other innate-like T cells, such as γδ T cells and NKT cells. GFP expression in Nur77 reporter mice is transient with a peak at 12–24 h after activation[45]. Therefore, it is likely that we underestimated the frequency of MAIT cells activated via their TCR in EAE. The fact that we observed the highest Nur77GFP signal in MAIT cells in the CNS already during the preclinical phase of EAE, might implicate that TCR activation occurs in

the periphery previous to MAIT cell infiltration into the inflamed CNS, where the Nur77 signal declines during the acute and chronic phase. However, we could not detect Nur77 induction by MAIT cells in lymph nodes during EAE. Peripheral cognate activation could still take place in other organs and could be mediated by antigens of procaryotic origin, most likely microbiome-derived, or even by MAIT-specific autoantigens, such as the recently identified sulfated bile acid, CA7S[10]. We can also not exclude local antigen-specific reactivation of MAIT cells in the CNS that could be mediated by bacterial antigens or CNS-derived autoantigens. The existence of a brain-specific microbiome was recently postulated as "BrainBiota"[50]. The antigen-binding groove of MR1 is not fully occupied by vitamin B metabolites[8,51] and MR1 is able to present endogenous or tumor antigens[52,53]. Furthermore, the fact that mature MAIT cells can also be detected in germ-free mice, can be interpreted as evidence for more unidentified self-antigens[54,55]. Such self-antigens could potentially serve as danger signals of tissue damage skewing innate-like MAIT cells towards a tissue repair phenotype.

CNS-infiltrating MAIT cells were almost all RORγt-positive with an increased frequency co-expressing T-bet in EAE. The formation of such double-positive MAIT1/17 cells has previously been observed in inflamed lungs of *Salmonella*- or *Legionella*-infected mice[13,35]. We here report their presence in the CNS and their enrichment in an auto-immunity model. There is evidence that the induction of MAIT1, MAIT17 and MAIT1/17 cells is associated with differential activation pathways, while the exact mechanisms are not fully understood[55]. It has been shown that TCR stimulation of MAIT cells rather leads to the induction of a type-17 phenotype[21], and that MAIT17 cells from the thymus express increased Nur77 levels compared to MAIT1 cells[54]. In contrast, TCR-independent activation via cytokines rather induces a type-1 phenotype. Human MAIT cells induce T-bet and IFN-γ after in vitro stimulation with IL-12 and IL-18[21]. IL-23 co-stimulates TCR-mediated MAIT cell activation[35] and in relation with IL-12 determines the MAIT1-MAIT17-ratio in a bacterial infection model[56]. Both, IL-12 and IL-23, are elevated in MS and EAE[57,58] and likely contribute to MAIT cell activation and phenotype differentiation in the inflamed CNS.

MAIT cells from the inflamed CNS possess features of an inflammatory pathogenic Th17-like as well as of a tissue repair phenotype. The cytokines IL-17A and GM-CSF contribute to angiogenesis, wound healing and barrier integrity in steady state[59,60]. However, in the context of autoimmunity it has been shown that IL-17A attracts neutrophils to the inflamed brain[61] and induces tertiary lymphoid tissue-like structures within the meninges further promoting CNS inflammation[62]. GM-CSF is the only so far identified inflammatory cytokine being indispensable for EAE induction in C57BL/6 J mice[63]. Mechanistically, GM-CSF promotes activation of microglia in EAE and increases monocyte migration into the CNS as well as their activation and differentiation[64,65]. Notably, IL-22 can either promote inflammation or enhance tissue repair and barrier integrity[66]. IL-22 has been associated with the pathogenicity of Th17 cells, whereas more recently a protective role of IL-22 in EAE has been shown by IL-22 overexpression[40]. In an EAU model, IL-22 produced by MAIT cells was protective[47]. Another potentially protective effector molecule upregulated by MAIT cells in the CNS in EAE is AREG. Here, we found TCR-mediated MAIT cell activation in the inflamed CNS to be associated with an increase in AREG production and a decrease in IL-17 production. Notably, we could further enhance AREG production by CNS-infiltrating MAIT cells in EAE by 5-OP-RU treatment. The treatment also led to overall increased amounts of AREG in the inflamed spinal cord. The latter might be amplified by secondary effects on other AREG sources, although we did not detect induction of AREG in other T cells or microglia in the CNS after treatment. Hence, our results establish MAIT cells as a potent source of AREG in EAE which can be therapeutically modulated by targeting MAIT cell activation. It was recently reported that AREG produced by MAIT cells increased wound healing

in the skin[67]. In the CNS, AREG produced by Tregs suppresses astrogliosis and thereby mediates neural recovery in an ischemic stroke model[41]. In EAE, microglia-derived AREG dampens astrocyte pro-inflammatory responses[42]. We observed decreased GFAP-expressing reactive astrocytes after 5-OP-RU treatment. Reactive astrocytes can directly contribute to tissue damage in the CNS and enhance immune cell activation and infiltration. Accordingly, the GFAP signal in astrocytes has been shown to correlate with disease severity in EAE[68]. Therefore, decreased GFAP after 5-OP-RU treatment in EAE further corroborates the therapeutic potential of TCR activation of MAIT cells in EAE.

Meningeal MAIT cells limit reactive oxygen species (ROS) and maintain blood-brain barrier (BBB) integrity thereby limiting neuroinflammation and preserving cognitive function in steady state[69]. However, this report showed that the expression of antioxidant molecules was decreased after MAIT cell activation, questioning the likelihood that such mechanisms also apply for strongly activated CNS-infiltrating MAIT cells in EAE. In addition, we could neither detect an increase in frequency nor TCR activation of meningeal MAIT cells in acute EAE. Together, there are multiple potential mechanisms that could mediate the protective effects of MAIT cells in EAE in addition to AREG, which require further specific in-depth investigation.

Here, we show that in EAE in vivo protective effector functions of MAIT cells enhanced by TCR-mediated activation outweigh their pathogenic potential. In line with published results[32], *Mr1*^−/− mice had an exacerbated EAE course. However, these mice exhibit a developmental phenotype, including increased intestinal barrier permeability resulting in more bacterial translocation[18]. Furthermore, the gut microbiome composition of *Mr1*^−/− mice differs from wildtype mice[33], which can influence immune cell maturation, the integrity of the BBB and neuronal pathways[70]. Thus, the altered EAE course in these mice is likely also influenced by these indirect developmental effects. In the same study, the authors also transferred T cells from a Vα19i-transgenic mouse, harboring an increased frequency of T cells expressing the same invariant TCR Vα chain as MAIT cells, and observed an amelioration of EAE. However, since the development of MR1-tetramers for identification of MAIT cells in wildtype mice, it is known that these Vα19i-transgenic T cells are phenotypically and functionally distinct and that results obtained with this mouse model need to be interpreted with caution[16,71]. It has previously been postulated that TCR-mediated activation induces a tissue repair phenotype in MAIT cells[21–23]. Here, we show that MAIT cells were activated via their TCR in EAE and that TCR-activated MAIT cells from the CNS produced more AREG and less IL-17A reflecting a more protective phenotype. Specific manipulation of the TCR-mediated activation of MAIT cells with an anti-MR1 antibody or their cognate antigen 5-OP-RU corroborated the protective effect of TCR-mediated MAIT cell activation in EAE. By this approach, we also avoided indirect developmental effects of *Mr1* gene deletion. While protective effects of TCR-mediated MAIT cell activation at low doses of 5-OP-RU became only evident in the chronic phase of the disease, high doses were already effective in the acute and recovery phase. It is possible that at low-dose treatment the ameliorating effect is overridden by the strong inflammatory signals from other T cells during the acute phase of the disease.

Although our data from the EAE model supports a protective function of MAIT cells in sterile CNS inflammation despite their pro-inflammatory potential, the question of translatability to MS remains to be addressed. However, antigen-specific MAIT cell activation holds the promise to be protective irrespective of their a priori function in MS, as it seems to skew MAIT cells towards a tissue repair and anti-inflammatory phenotype. In people with MS, MAIT cells have been detected in inflamed brain lesions[29–31], while their functional properties in the CNS are unknown. Having a model system of enriched MAIT cells in the CNS during EAE, we were able to circumvent many of the experimental limitations in humans and to include direct manipulation

to further decipher the role of MAIT cells in neuroinflammation. Linking TCR-mediated activation with tissue repair and protective function of MAIT cells in EAE renders them an accessible target for intervention in MS. MR1 and 5-OP-RU recognition by MAIT cells are evolutionarily highly conserved[72]. In addition, MAIT cells are even much more abundant in humans representing 1–10% of all blood T cells and up to 45% of liver T cells[12,17]. Effects of MAIT cell targeting can therefore be expected to be even stronger in humans. Knowing the specific antigen of MAIT cells in contrast to other T cell populations, holds the promise that antigen-specific MAIT cell activation could be protective and promote repair without directly affecting the function of other immune cells. 5-OP-RU might even be orally available[54] or an increased level could potentially be achieved by manipulation of the gut microbiome. Such a treatment approach could be tested not only in MS but also in other autoimmune diseases.

## Methods

### Mice
C57BL/6 J wildtype (The Jackson Laboratory, Charles River), B6.129P2-Mr1tm1Gfn ($Mr1^{-/-}$)[7] (provided by Olivier Lantz, Paris, France), B6(Cg)-Maithi Rorctm2Litt (RORγtGFP transgenic reporter mice)[20] (provided by Olivier Lantz, Paris, France) and B6N.B6-Tg(Nr4a1-EGFP/cre)820Khog/J (Nur77GFP reporter mice, strain #018974 from the Jackson Laboratory) were housed and bred under specific pathogen-free conditions in the animal facility of the University Medical Centre Hamburg-Eppendorf. For experiments evaluating the effects of MAIT TCR modulation on EAE, 6–7-week-old female C57BL/6 J wildtype mice were purchased from Charles River and acclimated to the new environment for at least 2 weeks before EAE induction. All mice were kept in a facility using a 12 h light/ 12 h dark cycle at temperatures of 20–25 °C with 40–70% humidity. Food and water were provided $ad$ $libitum$. We used adult mice (8–12 weeks old) from both sexes. For all experiments, littermates were used as control animals and mice from different experimental groups were mixed within cages to minimize cage-specific effects. Euthanasia was performed by using $CO_2$ followed by cardiac puncture (PBS perfusion) or cervical dislocation. All animal experimental procedures were in accordance to international and national animal welfare guidelines. Ethical approvals were obtained from the State Authority of Hamburg, Germany (approval no. 79/16 and 120/21).

### EAE
Mice were immunized subcutaneously with 200 µg of MOG$_{35-55}$ peptide (peptide and elephants, EP02030) in complete Freund's adjuvant (CFA, BD Biosciences, 263910) containing 2 mg/ml $Mycobacterium$ $tuberculosis$ (BD Biosciences, 231141). In addition, 200 ng pertussis toxin (Merck Millipore, 516560) in PBS were injected intraperitoneally (i.p.) on the day of immunization and 48 h later.

For adoptive transfer EAE, donor C57BL/6 J mice were actively immunized as described above. Nine days post immunization, single cell suspensions from LN (axillary, brachial and inguinal) and spleen were cultured at 37 °C and 5% $CO_2$ in cell culture medium (RPMI-1640, PAN Biotech, P04-18500; 10% FCS, Merck Millipore, F7524, Lot:BCBW9645; 1 mM sodium pyruvate, Thermo Fisher Scientific, 11360039; 1% GlutaMAX 100x, Thermo Fisher Scientific, 35050061; 1% HEPES 1 M pH 7.2–7.5, Capricorn, HEP-B; 1% non-essential amino acids 100x, Thermo Fisher Scientific, 11140050; 100 U/ml (100 µg/ml) penicillin and streptomycin, Thermo Fisher Scientific, 15140122; 50 µM β-mercaptoethanol, Thermo Fisher Scientific, 31350010) containing MOG$_{35-55}$ peptide (2 mg/ml, peptides and elephants, EP02030), IL-12 (100 ng/ml, PeproTech, 210-12) and anti-IFN-γ antibody (1.68 mg/ml, XMG1.2, BioLegend, 505858). After 72 h, the cells were harvested and CD4$^+$ T cells were enriched using the MojoSort CD4 T cell isolation kit (BioLegend, 480006) according to the manufacturer's instructions. $2 \times 10^6$ cells in PBS were intravenously injected in every recipient

mouse (C57BL/6 J). In addition, the recipient mice received pertussis toxin injections (200 ng in PBS, i.p.) at the day of cell injection and 2 days later.

Weight and clinical signs of disease were scored daily starting from day 7 by the following system: 0, no clinical deficits; 1, tail weakness; 2, hind limb paresis; 3, partial hind limb paralysis; 3.5, full hind limb paralysis; 4, full hind limb paralysis and forelimb paresis; 5, premorbid or dead. Animals reaching a clinical score of ≥ 4 or having more than 25% body weight loss (from starting weight) were euthanized according to regulations of the local Animal Welfare Act. EAE course experiments were scored blinded for the genotype or the respective treatment. Littermates were used as control animals and mice from different experimental groups were mixed within cages to minimize cage-specific effects. Missing values due to animals that were euthanized according to the regulations of the local Animal Welfare Act were imputed per mean clinical score per day of the respective group.

### EAE treatments
To generate 5-OP-RU from 5-A-RU, methylglyoxal solution (5 M, Merck Millipore, 67028) was diluted 1:100 with DMSO. 5 volumes of the methylglyoxal mix (500 mM) were incubated with 3 volumes of 5-A-RU (3.6 mM) for 48 hours at room temperature (RT) in darkness to generate 5-OP-RU. For low-dose in vivo application, 5-OP-RU was diluted with PBS to a concentration of 10 nmol/ml right before injection. Each mouse received an i.p. injection of 100 µl 5-OP-RU solution (10 nmol/ml) every 2 days starting at EAE onset (9 days post immunization). For high-dose in vivo application, the prodrug 5-A-RU-PABC-Val-Cit-Fmoc[46] (Biozol, CMS-CS-0132224) was used and diluted with PBS to a concentration of 300 nmol/ml right before injection. Each mouse received an i.p. injection of 100 µl 5-A-RU-PABC-Val-Cit-Fmoc solution (300 nmol/ml) for 5 days starting at EAE onset (9 days post immunization).

For blockade of MR1 during EAE, mice received i.p. injections 5, 10 and 15 days post immunization of either the anti-MR1 antibody (250 µg/injection, BioLegend, clone 26.5, 361110) or the respective IgG isotype control (250 µg/injection, BioLegend, clone MOPC-173, 400281) in PBS.

### Immune cell isolation
LN (inguinal, brachial and axillary) and spleen were homogenized through a 70 µm cell strainer and washed with PBS (300 g, 10 min, 4 °C). Cells from the spleen were resuspended in erylysis buffer (10 mM potassium bicarbonate, Merck Millipore, 237205; 0.15 M ammonium chloride, Merck Millipore, 213330; 0.1 mM Na$_2$EDTA, Thermo Fisher Scientific, 15576028; in ddH$_2$O; pH 7.3–7.4) and incubated for 5 min on ice. Lysis of red blood cells was stopped with PBS.

Mice were intracardially perfused with 10 ml PBS before the liver was taken and minced with a razor blade. After digestion (1 mg/ml collagenase D, Roche, 11088882001; 0.1 mg/ml DNase I, Merck Millipore, 260913; in RPMI-1640 medium, PAN Biotech, P04-18500) for 30 min at 37 °C in a shaking water bath, the tissue was homogenized through a 70 µm cell strainer. After washing with PBS, erythrocytes were lysed and remaining cells were resuspended in a 36% percoll solution (1.13 g/ml, GE Healthcare, 17089101) in RPMI-1640 medium and centrifuged for 20 min (800 × g, 4 °C) to enrich immune cells.

Mice were intracardially perfused with 10 ml PBS before the CNS was taken and minced with a razor blade. The tissue was digested for 45 min at 37 °C in a shaking water bath (1 mg/ml Collagenase A, Roche, 11088793001; 0.1 mg/ml DNase I, Merck Millipore, 260913; in RPMI-1640 medium) and subsequently homogenized through a 70 µm cell strainer. After washing with PBS, immune cells were isolated by percoll gradient centrifugation (30%/78% 1.13 g/ml, GE Healthcare, 17089101) at 2500 rpm, 30 min, 4 °C, w/o brake, harvested from the interphase and washed twice with PBS.

Mice were intracardially perfused with 10 ml PBS before the dural meninges were taken from the skull and minced with scissors. The

tissue was digested for 15 min at 37 °C in a shaking water bath (2 mg/ml Collagenase VIII, Merck Millipore, C2139; 0.2 mg/ml DNase I, Merck Millipore, 260913; in RPMI-1640 medium) and subsequently homogenized through a 70 μm cell strainer and washed with RPMI medium with 10% FCS.

## Flow cytometry

Single cell suspensions were incubated for 30 min at RT and 30 min at 4 °C with MR1- or CD1d-tetramers at concentrations between 1:300 and 1:800. Surface antigens were stained for 30 min at 4 °C with respective fluorochrome-coupled antibodies from BioLegend against CD3ε (1:100, 145-2C11, 100306), CD4 (1:100, GK1.5, 100447, 100453), CD8 (1:200, 53-6.7, 100750, 100759), CD11b (1:100, M1/70, 101228), CD11c (1:100, N418, 117318), CD44 (1:200, IM7, 103012, 103032) CD45 (1:200, 30-F11, 103116, 103128), CD45R (1:100, RA3-6B2, 103248), CD69 (1:100, H1.2F3, 104512), CD317 (1:100, 927, 127016), Ly6G (1:100, 1A8, 127624), PD-1 (1:100, 29 F.1A12, 135221), NK1.1 (1:100, PK136, 108708), TCR-β (1:100, H57-597, 109230), TCR-Vβ6 (1:100, RR4-7, 140006) or from BD Biosciences against CD11b (1:100, M1/70, 563553), CD19 (1:100, 1D3, 612971), CD45R (1:100, RA3-6B2, 612972), F4/80 (1:100, T45-2342, 565411), MHCII (1:100, M5/114.15.2, 563414), TCR-β (1:100, H57-597, 612821), TCR-Vβ8 (1:100, F23.1, 742378), TCR-γδ (1:100, GL3, 748989). Dead cells were stained using fixable viability stain 700 (BD Biosciences, 564997) for 20 min at 4 °C.

For intracellular staining, fixation and permeabilization of single cell suspensions was performed according to the manufacturer's protocol (BioLegend, fixation buffer, 420801; intracellular staining perm wash buffer, 421002) and stained with labelled antibodies against IL-22 (1:50, 1H8PWSR, eBioscience, 46-7222-80), GM-CSF (1:50, MP1-22E9, BioLegend, 505412), IFN-γ (1:50, XMG1.2, BioLegend, 505838), IL-17A (1:50, TC11-18H10.1, BioLegend, 506916), AREG (1:00, R&D Systems, AF989,) for 60 min at 4 °C. Unlabeled AREG antibody was conjugated to APC using the APC Conjugation Kit – Lightning-Link (abcam, ab201807) according to the manufacturer's instructions.

For intranuclear staining, cells endogenously expressed GFP were fixed with a 3% paraformaldehyde (PFA) solution for 60 min at RT to protect the GFP signal. Afterwards, the true-nuclear transcription factor buffer set (BioLegend, 424401) was used according to the manufacturer's instructions. Briefly, cells were fixed for 60 min at RT followed by two washing steps with permeabilization buffer and antibody staining against T-bet (1:50, 4B10, BioLegend, 644803) for 60 min at RT.

Absolute cell counts were quantified by using TruCount absolute counting tubes (BD Biosciences, 663028). All samples were acquired on a BD FACS LSR II analyzer or FACSymphony A3 (BD Biosciences). Flow cytometry-based cell sorting was performed on a FACSAria III cell sorter (BD Biosciences). Data was analyzed with FlowJo software (BD Biosciences, version 10.8).

## Tetramer preparation

Biotinylated MR1-5-OP-RU, MR1-6-FP, CD1d-PBS-57 and empty CD1d monomers (2 mg/ml) were obtained from the National Institutes of Health (NIH), aliquoted and stored at −80 °C. For tetramerization, monomers were incubated at a 4:1 molar ratio with fluorochrome labeled streptavidin (streptavidin-phycoerythrin, Thermo Fisher Scientific, S866 and streptavidin-BV421, BioLegend, 405225). Streptavidin was added stepwise at RT and tetramers were stored at 4 °C. The working concentration varied between 1:300 and 1:800 in different tetramerization approaches. In order to minimize staining differences, newly prepared tetramers were tested against previous preparations on LN, spleen or liver cells.

## T cell activation in vitro

Isolated immune cells were resuspended in cell culture medium (see above) and activated in different conditions at 37 °C and 5% CO₂.

PMA and ionomycin: 4 hours in the presence of phorbol 12-myristate-13-acetate (PMA) (10 ng/ml, Merck, P1585), ionomycin (1 μg/ml, Merck, 407950) and monensin (2 μM, BioLegend, 420701) at a density of 1 million cells per well in a 48 well plate.

Anti-CD3 and anti-CD28: 3 days in 96 well plates at a density of 250,000 cells per well with anti-CD3 (0.125 μg/ml, 145-2C11, BioLegend, 100314) and anti-CD28 (0.25 μg/ml, 37.51, BioLegend, 102116).

5-OP-RU: 2 (Supplementary Fig. 4g, h) or 3 (Supplementary Fig. 4f) days in 96 well plates at a density of 250 000 cells per well. 5-OP-RU was prepared as described above.

IL-12 and IL-18: mIL-12 (10 ng/ml, PeproTech, 210-12) and mIL-18 (12.5 ng/ml, Biozol, MBL-B002-5) was added for 2 days in 96 well plates at a density of 250 000 cells per well.

## Enzyme-linked immunosorbent assay (ELISA) of amphiregulin

Cervical spinal cords of EAE mice were homogenized using a tissue grinder in 2 mL radioimmuno-precipitation buffer (50 mM Tris, 150 mM NaCl, 0.5 mM EDTA, 10% SDS, 1% NP-40, 10% sodium deoxycholate, protease and phosphate inhibitor cocktails (cOmplete, Roche, 11836170001)), incubated at 4 °C for 30 min on a rotating wheel, and centrifuged for 5 min to remove the cell debris. After measuring the protein concentrations by a BCA assay (Pierce BCA Protein Assay Kit A (23228) and B (23224), Thermo Fisher Scientific) according to the manufacturer's protocol we used 50 μg protein in 10 μL lysate for the ELISA and measured technical duplicates. We used the DuoSet ELISA Mouse Amphiregulin kit (biotechne, DY989) and the DuoSet Ancillary Reagent Kit 2 (biotechne, DY008B) according to the manufacturer's protocol. The standard curve was generated using the provided recombinant protein where we measured a linear range until 500 pg/mL which was used to calculate the AREG amount in ng per μL spinal cord lysate. No technical replicates were excluded.

## Immunohistopathology

EAE mice were intracardially perfused with 4% PFA and the cervical spinal cords were dissected and fixed in 4% PFA for 1 hour at RT followed by dehydration in 30% sucrose solution in PBS for 1–2 days at 4 °C. Subsequently, the tissue was frozen in embedding solution (Tissue-Tek O.C.T compound) and cut into 12 μm thick transversal cryosections. Staining and imaging was performed at the UKE Mouse Pathology Facility. Slides were stained with hematoxylin (blue color) for orientation, followed by immunolabeling with an anti-GFAP antibody (M0761, Dako) that was visualized using avidin-biotin complex technique with 3,3'-diaminobenzidine (brown stain). Slides were analyzed with a NanoZoomer 2.0-RS digital slide scanner and NDP.view2 software (Hamamatsu). Area of GFAP signal was quantified with a customized mask using Fiji (ImageJ, version 2.0.0). Three images were analyzed per animal and the mean per animal was used for subsequent statistical comparisons.

## Bulk RNA sequencing of MAIT cells

Immune cells from the spleen of healthy C57BL/6 J mice and from spleen and CNS (n = 4) during acute EAE (14 days post immunization, score ≥ 3, active immunization protocol) were prepared as described above. Five mice were pooled per sample. Subsequently, T cells from the spleen were enriched by magnetic-activated cell sorting (MACS) using the Pan T cell isolation kit II (Miltenyi Biotec, 130-095-130) according to the manufacturer's instructions. Living CD45⁺CD11b⁻CD45R⁻TCR-β⁺MR1tetramer(5-OP-RU)⁺ MAIT cells were sorted using a FACSAria III cell sorter (BD Biosciences). RNA isolation was performed with the RNeasy Micro Kit (Qiagen, 74004) according to the manufacturer's instructions. RNA sequencing was performed at the next-generation sequencing (NGS) integrative genomics core unit (NIG) in Göttingen. RNA-sequencing libraries were prepared using the NEBNext Ultra RNA Library Prep Kit for Illumina (New England Biolabs)

with minor modifications in ligation and amplification and pooled and sequenced on a HiSeq 4000 sequencer (Illumina) generating 50 base pair single-end reads. Raw sequencing reads were aligned to the Ensembl mouse reference genome (GRCm38) using the aligner STAR (version 2.5) with default settings[73]. The overlap with annotated gene loci was counted with featureCounts. Further analyses were performed in the R environment (v.4.2.3) using publicly available packages. Differential expression analysis (Supplementary Data 1) was performed with DESeq2 (v.1.36.0) defining genes with a minimal twofold change and false discovery rate (FDR)-adjusted $P < 0.05$ differentially expressed. Sample similarity was assessed by using a principal component analysis (PCA) generated from normalized expression values after variance stabilizing transformation in DESeq2 using the top 500 most variable genes. The Gene Ontology (GO) analysis of biological process terms (Supplementary Data 2) was performed using gene set enrichment analysis within the R package clusterProfiler (v4.4.4)[74]. Input signatures were generated by ranking all expressed genes by the DESeq2-derived t statistics. Further gene set enrichment analyses were performed against custom gene sets (Supplementary Data 3 and Supplementary Data 4) derived from the literature, including MAIT cell activation[22], tissue repair[36,37], and pathogenic Th17 cells[4] (GSE39820, T16_60hr versus T36_60hr). Furthermore, we generated TCR exclusive and cytokine exclusive gene lists by removing the overlap between the TCR activation and cytokine activation gene sets[22]. To quantify the activity of these custom gene sets across samples we used AUCell (v.1.18.1)[75]. This method allows the computation of gene set activity values for individual samples or single cells (Supplementary Data 5) and thereby enables statistical comparison across experimental conditions while retaining biological replicates.

## Statistical analysis

Bar graphs represent mean ± standard error of the mean (SEM). Statistical analyses were performed using Prism software version 9.3.1 (GraphPad Software). Normal distribution was tested using Kolmogorov-Smirnov or Shapiro-Wilk test. Differences between two experimental groups were determined by Mann-Whitney-U test or Student's t-test. More than two groups were compared by one-way ANOVA with Bonferroni *post hoc* test. Two or more groups across multiple conditions were analyzed by two-way ANOVA with Bonferroni *post hoc* test. Significant results are indicated by *$P < 0.05$, **$P < 0.01$, ***$P < 0.001$.

## Reporting summary

Further information on research design is available in the Nature Portfolio Reporting Summary linked to this article.

## Data availability

Sequencing data generated for this study are available through the Gene Expression Omnibus under accession number GSE234291. All other data are included in the Supplementary Information/ Data and Source Data file. Source data are provided with this paper.

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

## Acknowledgements

We thank the UKE Central Animal Facility for their care and maintenance of our mouse lines and the UKE Mouse Pathology Facility for immuno-histopathology of spinal cord tissue from EAE mice. The MR1-5-OP-RU tetramer technology was developed jointly by Dr. James McCluskey, Dr. Jamie Rossjohn, and Dr. David Fairlie, and the material was produced by the NIH Tetramer Core Facility as permitted to be distributed by the University of Melbourne. This study was supported by the Deutsche Forschungsgemeinschaft (Grant No. 470154978; WI 5322/2-1 to A.W.) and Bundesministerium für Bildung und Forschung (Grant No. 01GI1605C to M.A.F.).

## Author contributions

M.W. conducted most of the experiments and analyzed the respective data. J.K.S. performed the adoptive transfer EAE. J.K.S., N.M., S.B., I.Winschel, M.S.W., I.Winkler and L.U. helped with mouse preparation and flow cytometry. G.S. did the RNA-sequencing. J.B.E. analyzed the RNA-sequencing data and performed all bioinformatic analyses. L.R. analyzed the immunohistopathology data. V.V. and M.S.W. performed and analyzed the AREG ELISA. O.L. contributed scientifically and pro-vided 5-A-RU and B6.129P2-Mr1$^{tm1Gfn}$ (*Mr1*$^{-/-}$) and B6(Cg)-Mait$^{hi}$Rorc$^{tm2Litt}$ mice. M.W., M.A.F., and A.W. designed experiments and wrote the manuscript. M.A.F. and A.W. supervised and funded the study.

## Funding

## Competing interests

The authors declare no competing interests.
