## [Transparent Peer Review file · Nature Communications]

Protective effect of TCR-mediated MAIT cell activation during experimental autoimmune encephalomyelitis

Corresponding Author: Professor Manuel Friese

Version 0:

Reviewer comments:

Reviewer #1

(Remarks to the Author)

This paper investigates the role of MAIT cells in mouse model of EAE. The authors characterized the cellular and molecular features of spleen and CNS MAIT cells in mouse model of EAE using flow cytometry analysis, RNA-seq, and QPCR. CNS MAIT cells were greatly increased in cellularity and had increased TCR activation in acute EAE. Compared with Spleen MAIT cells, CNS MAIT cells exhibited increased expression of multiple cytokines and growth factors including IL-22, Areg, IL17 and Csf2. Blockade of MR1/TCR signaling using MR1 blocking antibodies exacerbated clinical score of EAE; whereas stimulation of MR1/TCR signaling by 5-OP-RU reduced clinical score. While these data are interesting, there are a few concerns:

- (1) As the authors mentioned, a protective role for MAIT cells in mouse model of EAE has already been suggested by a previous study using V(alpha)19i transgenic mice. The novelty of the findings in the current study thus needs to be further clarified.
- (2) This paper yet lacks sufficient mechanistic insights in the current version. What are the specific molecular mechanisms by which MAIT cell activation leads to improved EAE symptom? Did treatments in Figures 5 and 6 influence such key molecular pathways in MAIT cells?
- (3) The authors provided interesting data that CNS MAIT cells were more activated compared to spleen MAIT cells in mouse model of EAE. The underlying mechanisms were not fully explored. How did CNS MAIT cells receive TCR signaling in EAE? Why did spleen MAIT cells not have similar responses?
- (4) The location of CNS MAIT cells is unclear. Are the CNS MAIT cells in the meninges? Or are these cells in the CNS parenchyma?

Reviewer #2

(Remarks to the Author)

The paper aims to dissect out the role of MAIT cells in EAE. There are opposite potential roles and here the authors focus ultimately on the later repair phase, where there seems to be a protective effect – interestingly one that can be boosted, which would be very relevant for therapy. The experiments seem to be well done and presented clearly, although the model itself is complicated by the different phases where MAIT depletion has opposite effects.

On the limitations side, overall, they haven't yet clarified what MAIT cells are doing in EAE and how they are having their protective role. They show that MAIT cell frequency is significantly increased in CNS (which is interesting) and that they are producing a range of effector and tissue repair molecules, but not which of these molecules are actually important for their function. The latter is a complex experiment due to the lack of cell specific knockouts, but was achieved in the study of the skin wounding model eg using adoptive transfers.

Specific comments

The text specifies that the pathogenic Th17 and two tissue repair signatures were specifically enriched in EAE CNS vs. EAE spleen MAIT cells, and not in either EAE spleen vs. healthy spleen or EAE CNS vs. healthy spleen (Fig. 2b,d). However, Supplementary Table 4 indicates a significant enrichment of these signatures in all three comparisons. Moreover, enrichment of the pathogenic Th17 signature is greater for the EAE spleen vs. healthy spleen comparison compared with the

EAE CNS vs. EAE spleen comparison. For the GSEA plots could the authors add p and FDR values on the figure rather than having to go hunting for them? (The legend in Figure 2 also needs attention)

The flow cytometry data for AREG expression in Fig. 3e,f is not very convincing. It suggests that almost all non-MAIT T cells and MAIT cells are expressing AREG in acute EAE. The staining pattern looks quite different from what is seen in Halgouet et al., *Immunity*, 2023 (Fig. 6e,f), although this is thymus and skin, and the shift between LN/spleen and CNS is minor. Healthy mouse samples should be included, so that the expression of AREG can be compared in healthy mice vs. EAE mice in the different tissues. Perhaps the staining approach used in the Halgouet et al., *Immunity*, 2023 paper could be used to show consistent results.

The experiment presented in Fig. 5 is lacking a control to show that i.p. administration of anti-MR1 reduces TCR-dependent MAIT cell activation.

Fig. 1 – is the presence of MAIT1/17 cells in CNS a feature of disease? Do MAIT1/17 cells express higher levels of CD69 compared with MAIT17 cells in CNS in acute EAE? What is the fraction of MAIT1/17 cells in healthy CNS (i.e. is the higher fraction of MAIT1/17 cells in EAE CNS vs. LN a feature of disease, or a feature of the CNS more generally)?

Fig. 2 – it would be useful to have RNA-seq data for MAIT cells in healthy CNS to understand how the phenotype of CNS MAIT cells changes with disease (in absence of confounding tissue differences).

Fig. 3 would be more informative if they examined the expression of effector and tissue repair molecules over the time course of EAE, given that they are suggesting the key role of MAIT cells in EAE is via tissue repair during the chronic phase. Does the functional profile of MAIT cells change during the course of EAE?

Fig. 3a – the expression of effector molecules by MAIT cells in EAE CNS is quite varied between mice. Does expression of any of these cytokines correlate with the severity of disease?

Fig. 3b – it would be useful to also show data for EAE spleen and healthy spleen, so that the protein results can be compared with the gene expression results in Fig. 3a.

Fig. 4a – given the large number of overlapping genes upregulated by TCR, cytokine, and TCR + cytokine stimuli, this analysis would be more informative if the gene signatures comprised genes specifically/more strongly upregulated in each condition. This could provide more insight into the dominant signalling pathway in EAE CNS.

The effect of altering MAIT cell activation (i.e. Fig. 5 and 6) on disease course is detectable but small and they used a large number of mice for these experiments relative to the other experiments to obtain statistical significance. For those not using EAE models it is hard to gauge what sort of impact these experiments illustrate and what the biological significance is. This could be addressed in the discussion – but also an experiment with an alternative intervention as a positive control would be very helpful to give context.

There are some inconsistencies in the model proposed in terms of at what stage MAIT cells are functioning. Fig. 5 and Fig. 6 indicate that MAIT cells are protective during the chronic phase, which they suggest is due to their tissue repair potential. However, MAIT cell frequency and activation peaks in the CNS during acute EAE, and Mr1^{-/-} mice have a higher clinical score at the acute phase (comparable clinical score to WT mice during chronic phase). It's also quite hard to reconcile the protective function of MAIT cells with their significant production of GM-CSF, which is pathogenic in EAE. This would warrant a little more discussion. The behaviour of MAIT cells is described as "ambivalent" but this sounds more like indecisive/mixed-up so maybe an alternative description is needed reflecting their distinct behaviours dependent on context/timing.

Reviewer #3

(Remarks to the Author)

Major findings of the study:

- MAIT cells (Va19Ja33, as defined by MR1-5-OP-RU binding CD45⁺TCR-b⁺ T cells) accumulated in the CNS in acute EAE (and declined again during the chronic phase of EAE). CNS MAIT cells had a transcriptome comprising effector molecules of pathogenic Th1/Th17 cells but also molecules associated with tissue repair, such as Areg.
- In order to disentangle whether proinflammatory functions or tissue repair functions of CNS MAIT cells prevailed, the authors looked at TCR engagement by staining for MAIT cells in Nur77 (GFP) reporter mice. CNS MAIT cells were Nur77⁺ with fast kinetics in the CNS (at onset) and AREG was co-expressed in Nur77⁺ rather than Nur77⁻ MAIT cells while it was not tested whether inflammatory cytokine expression also segregated with the Nur77 signal in CNS MAIT cells.
- A key finding of this study is that TCR engagement of MAIT cells might promote their tissue-regenerative function while cytokine activation of MAIT cells facilitates their pro-inflammatory effects. This idea had been suggested before by a series of laboratories but is linked to a relevant autoimmune situation in this study.
- Finally, blockade of MR-antigen presentation by an anti-MR antibody worsened chronic EAE while agonistic TCR activation of MAIT cells by 5-OP-RU that selectively targeted MAIT cells but not NKT cells ameliorated the chronic phase of

EAE, which was associated with less astrogliosis.

General critique:

This study draws attention to the increasingly well-characterized subset of MAIT cells in the context of an experimental autoimmune paradigm and extends previous studies to suggest that therapeutic interventions directed to the semi-invariant TCR of MAIT cells might be used to exploit an inherently tissue-reparative potential of MAIT cells in the CNS. This is an intriguing idea! While the study is overall well conducted, several inconsistencies in the line of argument need to be addressed in further experimentation to put the conclusions on a solid data basis and avoid misinterpretation.

Specific comments:

1. TCR engagement in MAIT cells is suggested to rather segregate with their tissue-regenerative function. This is an important statement since potential therapeutic interventions use the TCR-specificity of MAIT cells. The only confirmatory approach, that the authors take to link differential modes of MAIT cell activation with their functional phenotype is co-staining of Nur77 (GFP) with Areg. A more comprehensive analysis of this potentially highly relevant "dichotomy" is required to make a compelling argument for MAIT cell TCR-directed immunomodulatory interventions:

(i) Are Nur77 negative MAIT cells in the CNS pro-inflammatory?

(ii) A broader analysis of the transcriptome of Nur77 negative vs Nur77 positive MAIT cells would be required to support the authors' statement.

(iii) Linked to this: is the Nur77 reporter faithful enough to differentiate cytokine-driven from TCR-driven MAIT cells in the CNS?

(iv) Is the TCR-driven functional phenotype of MAIT cells uniform or is it possible that graded amounts of TCR engagement result in pro-inflammatory rather than regenerative functions? Here, a titration of the Nur77 signal in response to increasing amounts of MAIT TCR stimulation is shown but no functional outcome of the increasing TCR engagement.

2. The kinetics of Nur77 activation in CNS MAIT cells is highest in the preclinical phase of EAE and at disease onset while the fraction of TCR-activated MAIT cells (and also their absolute number) is minimal in the chronic phase of the disease. Yet, the clinical effects (loss and gain of function) are apparent only during the chronic phase of EAE (even though the effect sizes are not very convincing). How can this be reconciled with a truly regenerative function of MAIT cells? The data that are presented do not support such a statement.

3. Areg is used as a marker of "regenerative" MAIT cells. Is MAIT cell-derived Areg functionally relevant? - There are many alternative sources of Areg at the site of inflammation.

Minor comments:

Legend to supplementary Fig 1(d-f): Here, an adoptive transfer model is reported. Yet in the figure legend it says ("12-15 days post immunization"). This should be corrected.

Version 1:

Reviewer comments:

Reviewer #1

(Remarks to the Author)

The new data have greatly strengthened the manuscript. The authors have appropriately addressed all of my comments.

Reviewer #2

(Remarks to the Author)

This revised version of the manuscript addresses all the points raised very thoughtfully and the data presented are a nice addition to the paper. There are still plenty of open questions as to the source of the TCR trigger and the downstream mechanisms of the MAIT cells in the EAE model but overall they have clarified things very well.

Reviewer #3

(Remarks to the Author)

Even though my comments were not directly addressed, I appreciate that the authors performed a new in vivo experiment with 5-OU-RU, this time administered at the onset of EAE, which again resulted in an amelioration of EAE. By correlation, the authors now show that MAIT cell-associated AREG but not microglia- or T cell-associated AREG is increased upon treatment with 5-OU-RU. These data indicate that MAIT-TCR agonist treatment is on target. However, this experiment is no proof that AREG is the relevant effector molecule of MAIT cells to promote reduced inflammation/repair. The authors now discuss this appropriately in the revised discussion.

I still believe that the concept of this manuscript, i.e., agonist-induced therapeutic (and perhaps even regenerative) potential of MAIT cells, is interesting and might hold. However, the authors have not really addressed my concerns about potential

opposing functional phenotypes of TCR-triggered vs inflammation-triggered MAIT cells. They said it would be beyond the scope of the manuscript. However, since the clinical treatment effects of 5-OU-RU are very small (and I agree with reviewer #2 that it is hard to gauge whether this is relevant), it would be helpful to at least have a better characterization of potentially opposing functional phenotypes of MAIT cells (which after all also produce copious amounts of IL-17 and GM-CSF).

Version 2:

Reviewer comments:

Reviewer #3

(Remarks to the Author)

The authors have now performed a new experiment that is a key experiment to support their claim that TCR-triggered MAIT cells might be less inflammatory (or even have a protective phenotype) in autoimmune inflammation in the CNS. They sorted Nur77-positive vs. Nur77-negative MAIT cells from the inflamed CNS and interrogated these subsets for their functional phenotype. In particular, the fraction of AREG+ cells was significantly higher in the Nur77+ compartment than in the Nur77- compartment. Together with their evidence that treatment with 5-OP-RU is on target, these data support the idea that a TCR-targeted activation of MAIT cells might be exploited to modulate inflammation (and perhaps even facilitate regeneration) in the inflamed CNS.

POINT-BY-POINT RESPONSE TO REFEREES – NCOMMS-23-30482

The reviewers' comments are in *blue italics* and our responses are listed below each statement. Changes in the manuscript have been highlighted in **yellow**.

Reviewer #1 (expert in innate lymphocytes in the brain):

This paper investigates the role of MAIT cells in mouse model of EAE. The authors characterized the cellular and molecular features of spleen and CNS MAIT cells in mouse model of EAE using flow cytometry analysis, RNA-seq, and QPCR. CNS MAIT cells were greatly increased in cellularity and had increased TCR activation in acute EAE. Compared with Spleen MAIT cells, CNS MAIT cells exhibited increased expression of multiple cytokines and growth factors including IL-22, Areg, IL17 and Csf2. Blockade of MR1/TCR signaling using MR1 blocking antibodies exacerbated clinical score of EAE; whereas stimulation of MR1/TCR signaling by 5-OP-RU reduced clinical score. While these data are interesting, there are a few concerns:

Response: We were very pleased to read that this Reviewer found our data “interesting”. We thank this Reviewer for carefully assessing the manuscript that has helped us to further improve it.

(1) As the authors mentioned, a protective role for MAIT cells in mouse model of EAE has already been suggested by a previous study using V(alpha)19i transgenic mice. The novelty of the findings in the current study thus needs to be further clarified.

Response: The Reviewer is right that the transfer of T cells from a Vα19i-transgenic mouse led to an amelioration of EAE in a previous study (Croxford et al., Nat Immunol, 2006). However, since the development of MR1-tetramers for identification of MAIT cells in wildtype mice in 2015, it is known that these Vα19i-transgenic T cells are phenotypically and functionally distinct and that results obtained with this mouse model need to be interpreted with caution, as also argued by leading experts in the field of MAIT cell biology (Provine & Klenerman, Annu. Rev. Immunol. 38, 203–228 (2020)). Here, we used MR-1 tetramers to identify and characterize endogenous wildtype MAIT cells in EAE. We included a respective section in the discussion of the revised manuscript (**page 14**).

(2) This paper yet lacks sufficient mechanistic insights in the current version. What are the specific molecular mechanisms by which MAIT cell activation leads to improved EAE symptom? Did treatments in Figures 5 and 6 influence such key molecular pathways in MAIT cells?

Response: As requested, we performed additional experiments addressing the mechanism of action of EAE amelioration mediated by TCR specific MAIT cell activation. We can now show that 5-OP-RU treatment *in vivo* enhances AREG production by MAIT cells in the CNS and that AREG levels in the CNS tissue are increased in treated mice in acute EAE (**Fig. 7e, f**). Microglia-derived AREG has recently been reported to suppress disease-promoting astrocytes in EAE (Wheeler et al., Science, 2023). In line with this mechanism of action of AREG, we observe downstream ameliorating effects of TCR-mediated MAIT cell activation on astrogliosis (**Fig. 6f**). We believe that our data establish MAIT cells as an additional source of AREG in EAE which can be targeted in a specific therapeutic approach.

(3) The authors provided interesting data that CNS MAIT cells were more activated compared to spleen MAIT cells in mouse model of EAE. The underlying mechanisms were not fully explored. How did CNS MAIT cells receive TCR signaling in EAE? Why did spleen MAIT cells not have similar responses?

Response: In this study we specifically focused on the description and downstream effects of TCR-mediated MAIT cell activation in EAE. Although it is an intriguing question, we believe that extending our research also to the upstream mechanisms how MAIT cells get activated via their TCR in the EAE model, would be beyond the scope of the current study. The absence of signs of MAIT cell activation in the periphery (lymph nodes, **Suppl. Fig. 4f**) during EAE might indicate that TCR-mediated activation of MAIT cells takes place locally in the CNS. However, they might still first get activated in another peripheral organ and then home to the site of inflammation (CNS), as now more extensively discussed in the manuscript on **page 12**.

(4) The location of CNS MAIT cells is unclear. Are the CNS MAIT cells in the meninges? Or are these cells in the CNS parenchyma?

Response: As requested, we now quantified MAIT cells in the dural meninges showing that MAIT cells are also present there, but that their frequency does not change in acute EAE (**Suppl. Fig. 1b**). Furthermore, meningeal MAIT cells do not show signs of TCR activation in EAE to the same extent as CNS MAIT cells as quantified by Nur77 expression in the reporter mouse model (**Suppl. Fig. 4d, e**).

Reviewer #2 (expert in MAIT cells):

The paper aims to dissect out the role of MAIT cells in EAE. There are opposite potential roles and here the authors focus ultimately on the later repair phase, where there seems to be a protective effect – interestingly one that can be boosted, which would be very relevant for therapy. The experiments seem to be well done and presented clearly, although the model itself is complicated by the different phases where MAIT depletion has opposite effects. On the limitations side, overall, they haven't yet clarified what MAIT cells are doing in EAE and how they are having their protective role. They show that MAIT cell frequency is significantly increased in CNS (which is interesting) and that they are producing a range of effector and tissue repair molecules, but not which of these molecules are actually important for their function. The latter is a complex experiment due to the lack of cell specific knockouts, but was achieved in the study of the skin wounding model eg using adoptive transfers.

Response: We were pleased to read that the Reviewer found our experiments “well done and presented clearly”. The detailed and careful analysis of our study by this Reviewer has been very helpful to further improve our manuscript. By addressing the questions how MAIT cells exert their protective role and which molecule is important for EAE amelioration by TCR specific MAIT cell activation, we now show that 5-OP-RU treatment of EAE enhances AREG production by MAIT cells in the CNS, quantified with the adjusted flow cytometry protocol as requested in specific comment 2 (**Fig. 7e**). In addition, AREG levels in the CNS tissue were increased by 5-OU-RU treatment in acute EAE (**Fig. 7f**). Microglia-derived AREG has recently been reported to suppress disease-promoting astrocytes in EAE (Wheeler et al., Science, 2023). In line with this mechanism of action of AREG, we observe downstream ameliorating effects of TCR-mediated MAIT cell activation on astrogliosis (**Fig. 6f**).

Specific comments

1. The text specifies that the pathogenic Th17 and two tissue repair signatures were specifically enriched in EAE CNS vs. EAE spleen MAIT cells, and not in either EAE spleen vs. healthy spleen or EAE CNS vs. healthy spleen (Fig. 2b,d). However, Supplementary Table 4 indicates a significant enrichment of these signatures in all three comparisons. Moreover, enrichment of the pathogenic Th17 signature is greater for the EAE spleen vs. healthy spleen comparison compared with the EAE CNS vs. EAE spleen comparison. For the GSEA plots could the authors add p and FDR values on the figure rather than having to go hunting for them? (The legend in Figure 2 also needs attention)

Response: We thank the Reviewer for pointing out this inconsistency. We now extended the description of both analyses, GSEA and AUCell, on **page 7 and 8** of the manuscript. We would like to point out that the enrichment of all three gene sets can be considered most robust in CNS-infiltrating MAIT cells, as it was identified by two analysis methods, while the enrichment of the same gene sets in MAIT cells from the spleen in EAE compared to those from healthy spleen was only evident in GESEA but not AUCell analyses. As requested, we added normalized enrichment scores (NES) and FDR values (*P* values after Benjamini-Hochberg adjustment) to **Figure 2b and d** and **Suppl. Fig. 3c** and corrected the figure legend of figure 2.

2. The flow cytometry data for AREG expression in Fig. 3e,f is not very convincing. It suggests that almost all non-MAIT T cells and MAIT cells are expressing AREG in acute EAE. The staining pattern looks quite different from what is seen in Halgouet et al., Immunity, 2023 (Fig. 6e,f), although this is thymus and skin, and the shift between LN/spleen and CNS is minor. Healthy mouse samples should be included, so that the expression of AREG can be compared in healthy mice vs. EAE mice in the different tissues. Perhaps the staining approach used in the Halgouet et al., Immunity, 2023 paper could be used to show consistent results.

Response: As requested, we repeated the flow cytometry experiment using the same staining approach as Halgouet et al. and included EAE spleen and healthy spleen to allow consistent comparison of the protein results with the gene expression results (**Fig. 3e, f**). In line with the transcriptome data, AREG was significantly increased in CNS-infiltrating MAIT cells in EAE in comparison to peripheral MAIT cells.

3. The experiment presented in Fig. 5 is lacking a control to show that i.p. administration of anti-MR1 reduces TCR-dependent MAIT cell activation.

Response: We agree with the Reviewer that this is an important control experiment, which we now performed with Nur77GFP reporter mice showing decreased Nur77 expression, i.e. TCR activation, in CNS-infiltrating MAIT cells after administration of anti-MR1 following the same treatment regimen as used for comparison of EAE disease courses in **Fig. 5 a-c**. The effect was MAIT cell specific as there was no alteration of Nur77 expression in CNS infiltrating non-MAIT T cells in the same animals. The data is depicted in **Fig. 5d, e**.

4. Fig. 1 – is the presence of MAIT1/17 cells in CNS a feature of disease? Do MAIT1/17 cells express higher levels of CD69 compared with MAIT17 cells in CNS in acute EAE? What is the fraction of MAIT1/17 cells in healthy CNS (i.e. is the higher fraction of MAIT1/17 cells in EAE CNS vs. LN a feature of disease, or a feature of the CNS more generally)?

Response: We thank the Reviewer for raising this question, which we addressed by new flow cytometry analyses depicted in **Figure 1 g-i**. Indeed, we also found MAIT1/17 cells in the CNS of healthy mice, and their frequency was significantly increased in EAE (**Fig. 1g, h**). Therefore, in our view, these cells can be regarded as feature of the CNS as well as of EAE, where they also show stronger signs of activation in comparison to MAIT17 cells (**Fig. 1i**).

5.Fig. 2 – it would be useful to have RNA-seq data for MAIT cells in healthy CNS to understand how the phenotype of CNS MAIT cells changes with disease (in absence of confounding tissue differences).

Response: Indeed, it would be informative to have RNA-seq data of MAIT cells from the healthy CNS. However, this is confounded by the extremely low number of MAIT cells in this organ at steady state. Still, we made every possible effort and isolated 4 x 10⁵ – 200 MAIT cells from the CNS of in total 60 healthy mice to subject them to ultra-low input RNA sequencing. Although in all samples 28 million reads or more could be generated, only a small fraction of less than 10% in most samples could be assigned to genomic exons, leaving us with less than 2 million reads for most samples. Together with a sequence doublet rate of more than 80% for most samples, we would consider this data set of insufficient quality. This is most likely due to too low input material. In addition, the input quality might be compromised resulting from long sorting times due to the low frequency of MAIT cells within the immune cell isolates from the healthy CNS. Thus, despite having undertaken every possible effort to provide this data, we have to accept that the isolation of the few MAIT cells present in healthy CNS at high enough numbers for RNA sequencing appears to be technically extremely challenging. To still provide insight into the question whether the phenotype of CNS MAIT cells is a feature of disease or of the organ, we provide new flow cytometry data (see response to comment 4).

6.Fig. 3 would be more informative if they examined the expression of effector and tissue repair molecules over the time course of EAE, given that they are suggesting the key role of MAIT cells in EAE is via tissue repair during the chronic phase. Does the functional profile of MAIT cells change during the course of EAE?

Response: The flow cytometry data in figure 3 primarily serves as validation of the transcriptome data, which were acquired during the acute phase of EAE where inflammatory responses are the strongest. We agree with the Reviewer that additional analyses over the time course of EAE would have been required to address the time discrepancy of MAIT cell peak activation at disease onset and the effects of targeting TCR-mediated MAIT cell activation, which were only evident in the chronic phase. However, we now present new data treating EAE mice with higher doses of 5-OP-RU at disease onset, which was already effective shortly after treatment, during the acute and recovery phase (**Fig. 7a-c**). In light of this new result, it is possible that at low-dose treatment the ameliorating effect of TCR-mediated MAIT cell activation is overridden by the strong inflammatory signals from other T cells during the acute phase of the disease (see discussion **page 15**).

7.Fig. 3a – the expression of effector molecules by MAIT cells in EAE CNS is quite varied between mice. Does expression of any of these cytokines correlate with the severity of disease?

Response: We agree with the Reviewer that the variation in cytokine expression levels at RNA level depicted in **Fig. 3a** is high. However, the values do not correlate with the score of the animals analyzed (Spearman rank correlation, data not shown). We ensured a priori that the mean score of the animals pooled per sample ($n = 5$ animals per sample) for sequencing

was comparable, varying from 3.1 to 3.35, with individual animals having a score between 3.0 and 3.5.

8.Fig. 3b – it would be useful to also show data for EAE spleen and healthy spleen, so that the protein results can be compared with the gene expression results in Fig. 3a.

Response: As requested, we now also analyzed the expression of cytokines in MAIT cells from EAE spleen and healthy spleen at protein level by flow cytometry and included the data in **Figure 3b, c**. The results are in line with the transcriptome data, confirming the enhanced type 17 phenotype of CNS-infiltrating MAIT cells in EAE in comparison to peripheral MAIT cells.

9.Fig. 4a – given the large number of overlapping genes upregulated by TCR, cytokine, and TCR + cytokine stimuli, this analysis would be more informative if the gene signatures comprised genes specifically/more strongly upregulated in each condition. This could provide more insight into the dominant signalling pathway in EAE CNS.

Response: We agree with the Reviewer that a more detailed dissection of the TCR vs cytokine-induced signatures would be valuable. Therefore, we compared the TCR activation and cytokine activation gene lists from Leng *et al.*, 2019 and generated TCR-exclusive and cytokine-exclusive gene lists by subtracting genes regulated in both conditions (updated **Suppl. Table 3**). We tested the upregulation of these exclusive gene sets in our transcriptome data of MAIT cells via GSEA and AUCCell analyses. These analyses showed comparable results to our analyses with whole TCR activation and cytokine activation gene sets. The analyses are presented in **Suppl. Fig. 4b, c, d**.

10.The effect of altering MAIT cell activation (i.e. Fig. 5 and 6) on disease course is detectable but small and they used a large number of mice for these experiments relative to the other experiments to obtain statistical significance. For those not using EAE models it is hard to gauge what sort of impact these experiments illustrate and what the biological significance is. This could be addressed in the discussion – but also an experiment with an alternative intervention as a positive control would be very helpful to give context.

Response: We agree with the reviewer that the effects of MAIT cell inhibition and activation in Figures 5 and 6 are small. Therefore, we performed an additional experiment with an adjusted 5-OP-RU treatment regimen consisting of 5 consecutive daily injections of higher doses of 5-OP-RU (30 nmol / injection) at EAE onset. We chose to treat at such high dose only during the phase of acute inflammation and for a short time period to avoid MAIT cell depletion. Indeed, this regimen led to stronger effects on the EAE disease course which were then already evident shortly after treatment in the acute and recovery phase (**Fig. 7a-c**). Still, in our view, the effects of such a specific treatment targeting a single T cell sub-population cannot directly be compared to effects of treatments such as sphingosine-1-phosphate receptor modulators (i.e. Fingolimod) targeting all T cells, although this represents the gold standard positive control in drug development nowadays. Even if the effect of 5-OP-RU-mediated MAIT cell activation in mice is still moderate, even at high doses, it is to be expected that effects in humans would be much stronger as MAIT cells are 10 to 30-times more abundant in adult humans compared to laboratory mice, as discussed in the manuscript on **page 15**.

11.There are some inconsistencies in the model proposed in terms of at what stage MAIT cells are functioning. Fig. 5 and Fig. 6 indicate that MAIT cells are protective during the chronic phase, which they suggest is due to their tissue repair potential. However, MAIT cell

frequency and activation peaks in the CNS during acute EAE, and Mr1^{-/-} mice have a higher clinical score at the acute phase (comparable clinical score to WT mice during chronic phase). It's also quite hard to reconcile the protective function of MAIT cells with their significant production of GM-CSF, which is pathogenic in EAE. This would warrant a little more discussion. The behaviour of MAIT cells is described as "ambivalent" but this sounds more like indecisive/mixed-up so maybe an alternative description is needed reflecting their distinct behaviours dependent on context/timing.

Response: We thank the Reviewer for this thoughtful and critical comment, which led us to improve the wording and discussion of our manuscript. As already discussed in our response to comment 6 of this Reviewer, in light of our new results of a stronger and earlier effect of 5-OP-RU treatment at high doses (**Fig. 7a-c**), it is possible that the late effects of low-dose treatment are due to the strong inflammatory signals from other T cells overriding treatment effects during the acute phase of the disease. This would also be consistent with MAIT cell deficiency in *Mr1^{-/-}* mice exacerbating the acute phase of the disease. As requested, we adjusted and shaped the consistency of our description of MAIT cell phenotype and function in EAE throughout the manuscript. Furthermore, we added a specific section discussing the discrepancy between phenotypically protective and pro-inflammatory potential and the functionally protective effects of TCR-mediated MAIT cell activation in the context of translatability of our results (**page 15**).

Reviewer #3 (expert in T cells and CNS autoimmunity):

Major findings of the study:

- *MAIT cells (Va19Ja33, as defined by MR1-5-OP-RU binding CD45+TCR-b+ T cells) accumulated in the CNS in acute EAE (and declined again during the chronic phase of EAE). CNS MAIT cells had a transcriptome comprising effector molecules of pathogenic Th1/Th17 cells but also molecules associated with tissue repair, such as Areg.*
- *In order to disentangle whether proinflammatory functions or tissue repair functions of CNS MAIT cells prevailed, the authors looked at TCR engagement by staining for MAIT cells in Nur77 (GFP) reporter mice. CNS MAIT cells were Nur77+ with fast kinetics in the CNS (at onset) and AREG was co-expressed in Nur77+ rather than Nur77- MAIT cells while it was not tested whether inflammatory cytokine expression also segregated with the Nur77 signal in CNS MAIT cells.*
- *A key finding of this study is that TCR engagement of MAIT cells might promote their tissue-regenerative function while cytokine activation of MAIT cells facilitates their pro-inflammatory effects. This idea had been suggested before by a series of laboratories but is linked to a relevant autoimmune situation in this study.*
- *Finally, blockade of MR-antigen presentation by an anti-MR antibody worsened chronic EAE while agonistic TCR activation of MAIT cells by 5-OP-RU that selectively targeted MAIT cells but not NKT cells ameliorated the chronic phase of EAE, which was associated with less astrogliosis.*

General critique:

This study draws attention to the increasingly well-characterized subset of MAIT cells in the context of an experimental autoimmune paradigm and extends previous studies to suggest that therapeutic interventions directed to the semi-invariant TCR of MAIT cells might be used to exploit an inherently tissue-reparative potential of MAIT cells in the CNS. This is an intriguing idea! While the study is overall well conducted, several inconsistencies in the line

of argument need to be addressed in further experimentation to put the conclusions on a solid data basis and avoid misinterpretation.

Response: We were very pleased to read that this Reviewer found our study to be “overall well conducted”. We thank this Reviewer for carefully assessing the manuscript that encouraged us to address inconsistencies by new experiments and to adjust our line of argument, resulting in further improvement of the manuscript.

Specific comments:

1. TCR engagement in MAIT cells is suggested to rather segregate with their tissue-regenerative function. This is an important statement since potential therapeutic interventions use the TCR-specificity of MAIT cells. The only confirmatory approach, that the authors take to link differential modes of MAIT cell activation with their functional phenotype is co-staining of Nur77 (GFP) with Areg. A more comprehensive analysis of this potentially highly relevant "dichotomy" is required to make a compelling argument for MAIT cell TCR-directed immunomodulatory interventions:

Response: As requested by the Reviewer, we performed further experiments to consolidate the link between TCR-mediated MAIT cell activation and their protective phenotype and function in EAE. Most importantly, we now provide data that in vivo treatment with 5-OP-RU leads to increased expression of AREG specifically in MAIT cells in EAE and to increased levels of AREG in spinal cord lysates (**Fig. 7e, f**).

As suggested by Reviewer 2 (comment 2), we adjusted the AREG flow cytometry staining approach according to Halgouet et al., Immunity, 2023. To achieve reliable results, it is necessary to stimulate cells ex vivo with PMA/ionomycin in the presence of monensin, which excludes co-staining for Nur77.

However, we believe that the now provided direct link of TCR-specific MAIT cell activation in vivo with increased levels of AREG, as an established mediator of protective effects in EAE (Wheeler et al., Science, 2023), is compelling to support our statement that TCR-mediated MAIT cell activation is protective in EAE and that this effect is at least partially mediated by AREG. Furthermore, we observe downstream ameliorating effects of TCR-mediated MAIT cell activation on astrogliosis (**Fig. 6f**), confirming the mechanism of action of AREG in EAE as described by Wheeler et al. (Science, 2023).

(i) Are Nur77 negative MAIT cells in the CNS pro-inflammatory?

Response: We agree with the Reviewer that in addition to showing that TCR activation of MAIT cells exerts protective function, it would be of interest to investigate whether MAIT cells that have been activated in an antigen-independent manner (e.g. via cytokines such as IL-18 and IL-12) exert pro-inflammatory function. However, in this study we focus on the effects of TCR-mediated MAIT cell activation on EAE and now provide more data linking this therapeutic approach directly to AREG as a candidate effector molecule (see above). We believe that extending our research also to the potentially pro-inflammatory effects of cytokine-mediated MAIT cell activation, would be beyond the scope of the current study.

(ii) A broader analysis of the transcriptome of Nur77 negative vs Nur77 positive MAIT cells would be required to support the authors' statement.

Response: We thank the Reviewer for suggesting this valuable experimental approach. Although highly interesting, in this study we focus on the effects of TCR-mediated MAIT cell activation in CNS inflammation and use the Nur77 reporter as a tool to quantify TCR-

mediated MAIT cell activation. Here, we did not aim to contrast TCR- vs cytokine-activated MAIT cell subsets in general and adjusted our description of 5-OP-RU treatment effects throughout the manuscript and the discussion to clarify our approach.

(iii) Linked to this: is the Nur77 reporter faithful enough to differentiate cytokine-driven from TCR-driven MAIT cells in the CNS?

Response: As outlined above the focus of this study is on the effects of TCR-mediated activation of MAIT cells on CNS inflammation. In this context, we used the Nur77 reporter as a tool to quantify TCR-mediated MAIT cell activation. This method is established in the field and has been used by several other groups as reporter for TCR-mediated MAIT cell activation, as for example in Halgouet et al., *Immunity*, 2023 and Yamana et al. *Mucosal Immunology*, 2022. Confirming the feasibility in our model, we show that in vitro and in vivo treatment of MAIT cells with their cognate antigen 5-OP-RU led to induction of Nur77GFP specifically in MAIT cells and no other T cells and that this activation is dose dependent (**Fig. 6a, b**). Furthermore, in the transcriptome data published by Leng et al. (*Cell Rep*, 2019) Nr4a1 (Nur77) was upregulated in MAIT cells only after TCR stimulation and after stimulation via TCR + cytokines, but not by cytokine stimulation alone.

(iv) Is the TCR-driven functional phenotype of MAIT cells uniform or is it possible that graded amounts of TCR engagement result in pro-inflammatory rather than regenerative functions? Here, a titration of the Nur77 signal in response to increasing amounts of MAIT TCR stimulation is shown but no functional outcome of the increasing TCR engagement.

Response: Indeed, it would be interesting to disentangle the effects of graded amounts of TCR engagement on MAIT cell phenotype and function. In our in vivo experiments, both low and high doses of 5-OP-RU led to an amelioration of EAE clinical symptoms. As described above, co-staining of Nur77 and the functional effector molecule investigated in our study, AREG, is not possible in flow cytometry due to the necessity of PMA and ionomycin stimulation prior to AREG staining.

2. The kinetics of Nur77 activation in CNS MAIT cells is highest in the preclinical phase of EAE and at disease onset while the fraction of TCR-activated MAIT cells (and also their absolute number) is minimal in the chronic phase of the disease. Yet, the clinical effects (loss and gain of function) are apparent only during the chronic phase of EAE (even though the effect sizes are not very convincing). How can this be reconciled with a truly regenerative function of MAIT cells? The data that are presented do not support such a statement.

Response: To first address the Reviewers concern that the effect of TCR-mediated MAIT cell activation by 5-OP-RU on the EAE disease score is small, we conducted an additional experiment with an adjusted treatment regimen consisting of 5 consecutive daily injections of higher doses of 5-OP-RU (30 nmol / injection) during the acute phase of EAE. We chose to treat at such high dose only during the phase of acute inflammation and for a short time period to avoid MAIT cell depletion. Indeed, this regimen led to more pronounced effects on the EAE disease course which were then also already evident shortly after treatment in the acute and recovery phase (**Fig. 7a-c**). These results address the concern of discrepancy between peak TCR-activation of MAIT cells in untreated animals and the timepoint of treatment effects of low-dose 5-OP-RU. These findings indicate that in addition to supporting recovery, TCR-mediated MAIT cell activation can exert protective effects already during acute inflammation, in line with the recently reported function of microglial AREG as a suppressor of disease-promoting astrocytes in EAE (Wheeler et al., *Science*, 2023). In light of our new results, it is possible that at low-dose treatment, the ameliorating effect of TCR-

mediated MAIT cell activation on disease-promoting astrocyte functions via MAIT cell derived AREG is overridden by the strong inflammatory signals from other T cells during the acute phase of the disease (see discussion **page 15**). Regarding the true clinical relevance of MAIT cell protective and repair functions, we would like to point out that MAIT cells are 10 to 30-times more abundant in adult humans than in laboratory mice. Therefore, it can be expected that treatment effects in humans would be considerably stronger.

3. Areg is used as a marker of "regenerative" MAIT cells. Is MAIT cell-derived Areg functionally relevant? - There are many alternative sources of Areg at the site of inflammation.

Response: We agree with the Reviewer that MAIT cells are not the only source of AREG at the site of inflammation, the CNS in EAE. As cited in the manuscript Treg derived AREG has been shown to mediate neuronal recovery in an ischemic stroke model (Ito et al., Nature, 2019) and microglial AREG suppresses disease-promoting astrocyte functions in EAE (Wheeler et al., Science, 2023). However, we now provide new data showing that in vivo 5-OP-RU treatment, which is specifically targeting only MAIT cells, leads to increased AREG production by MAIT cells in EAE, while other T cells and microglia did not induce AREG (**Fig. 7e**). Moreover, levels of AREG in spinal cord lysates were increased in 5-OP-RU treated animals (**Fig. 7f**). We believe that our data establish MAIT cells as an additional source of AREG in EAE which can be targeted in a specific therapeutic approach.

Minor comments:

Legend to supplementary Fig 1(d-f): Here, an adoptive transfer model is reported. Yet in the figure legend it says ("12-15 days post immunization"). This should be corrected.

Response: We thank the Reviewer for pointing out this inaccuracy and have corrected the figure legend accordingly.

POINT-TO-POINT RESPONSE TO REFEREES – NCOMMS-23-30482A

The reviewers' comments are in *blue italics* and our responses are listed below each statement. Changes in the manuscript have been highlighted in **yellow**.

Reviewer #1 (Remarks to the Author):

The new data have greatly strengthened the manuscript. The authors have appropriately addressed all of my comments.

Response: We were very pleased to read that this Reviewer found that the "new data greatly strengthen the manuscript" and thank them again for their constructive feedback on the first version.

Reviewer #2 (Remarks to the Author):

This revised version of the manuscript addresses all the points raised very thoughtfully and the data presented are a nice addition to the paper. There are still plenty of open questions as to the source of the TCR trigger and the downstream mechanisms of the MAIT cells in the EAE model but overall they have clarified things very well.

Response: We were pleased to read that this Reviewer acknowledges our thoughtful first revision and thank them again for their constructive feedback. We agree that there are plenty of open questions on MAIT cells in EAE and MS to be tackled in the future.

Reviewer #3 (Remarks to the Author):

Even though my comments were not directly addressed, I appreciate that the authors performed a new in vivo experiment with 5-OU-RU, this time administered at the onset of EAE, which again resulted in an amelioration of EAE. By correlation, the authors now show that MAIT cell-associated AREG but not microglia- or T cell-associated AREG is increased upon treatment with 5-OU-RU. These data indicate that MAIT-TCR agonist treatment is on target. However, this experiment is no proof that AREG is the relevant effector molecule of MAIT cells to promote reduced inflammation/repair. The authors now discuss this appropriately in the revised discussion.

Response: We were pleased to read that the Reviewer appreciates our new experiments and the adapted discussion.

I still believe that the concept of this manuscript, i.e., agonist-induced therapeutic (and perhaps even regenerative) potential of MAIT cells, is interesting and might hold. However, the authors have not really addressed my concerns about potential opposing functional phenotypes of TCR-triggered vs inflammation-triggered MAIT cells. They said it would be beyond the scope of the manuscript. However, since the clinical treatment effects of 5-OU-RU are very small (and I agree with reviewer #2 that it is hard to gauge whether this is relevant), it would be helpful to at least

have a better characterization of potentially opposing functional phenotypes of MAIT cells (which after all also produce copious amounts of IL-17 and GM-CSF).

Response: As requested by the Reviewer we performed new *in vitro* and *in vivo* experiments to better characterize TCR-activated vs cytokine-activated MAIT cells. First, we compared Nur77, GM-CSF, IL-17 and AREG expression of MAIT cells after 5-OP-RU and/or IL-12 and IL-18 stimulation *in vitro* showing that IL-17 and AREG are induced by both TCR- and cytokine stimulation, while GM-CSF was only induced by cytokines. In the case of AREG, we could observe an additive effect of combining TCR and cytokine stimulation, which we did not observe for IL-17 (**Suppl. Fig. 4g, h**). To then better characterize the functional phenotype of TCR-triggered MAIT cells *in vivo* during EAE, we isolated Nur77-positive and Nur77-negative MAIT cells from the CNS in the acute phase of the disease, stimulated the cells *ex vivo* with PMA and ionomycin and quantified GM-CSF, IL-17 and AREG in the two subsets. Indeed Nur77GFP+ MAIT cells had an increased expression of AREG and decreased expression of IL-17, corroborating the protective potential of TCR-mediated MAIT cell activation in EAE. GM-CSF was not differentially expressed in TCR-activated Nur77+ MAIT cells (**Fig. 4 e,f**). We believe that this clearly illustrates the opposing functional phenotypes of MAIT cells.